# Concerted roles of LRRTM1 and SynCAM 1 in organizing prefrontal cortex synapses and cognitive functions

Karen Perez de Arce[1,6,10], Adema Ribic [1,7,10], Dhrubajyoti Chowdhury[2,10], Katherine Watters[1,2,10], Garth J. Thompson [3,8], Basavaraju G. Sanganahalli [3], Elizabeth T. C. Lippard [3,9], Astrid Rohlmann [4], Stephen M. Strittmatter [2,5], Markus Missler [4], Fahmeed Hyder [3] & Thomas Biederer [1,2] ✉

Multiple trans-synaptic complexes organize synapse development, yet their roles in the mature brain and cooperation remain unclear. We analyzed the postsynaptic adhesion protein LRRTM1 in the prefrontal cortex (PFC), a region relevant to cognition and disorders. LRRTM1 knockout (KO) mice had fewer synapses, and we asked whether other synapse organizers counteract further loss. This determined that the immunoglobulin family member SynCAM 1 controls synapse number in PFC and was upregulated upon LRRTM1 loss. Combined LRRTM1 and SynCAM 1 deletion substantially lowered dendritic spine number in PFC, but not hippocampus, more than the sum of single KO impairments. Their cooperation extended presynaptically, and puncta of Neurexins, LRRTM1 partners, were less abundant in double KO (DKO) PFC. Electrophysiology and fMRI demonstrated aberrant neuronal activity in DKO mice. Further, DKO mice were impaired in social interactions and cognitive tasks. Our results reveal concerted roles of LRRTM1 and SynCAM 1 across synaptic, network, and behavioral domains.

Synapse formation proceeds rapidly in early postnatal stages and continues as the brain matures[1,2]. Trans-synaptic complexes control synapse development and maturation[3–6], and a rich complement of adhesion proteins spans the synaptic cleft[7,8]. This diversity opens up a profound cooperative potential that is incompletely understood[9,10]. The question is important for disorders involving synaptopathies including schizophrenia, where mutations associated with synaptic processes and excitatory transmission convey risk[11,12].

Intriguingly, synapses are not affected across all brain areas in disorders, pointing to region-select vulnerabilities. An example is the prefrontal cortex (PFC), a region impacted in multiple brain disorders, including disrupted maturation and connectivity in schizophrenia with symptoms generally appearing in late adolescence[13–15]. Patients exhibit in the PFC layer III a lower number of dendritic spines, the postsynaptic protrusions onto which most excitatory synapses are formed[16,17]. Pyramidal neurons in the PFC are highly interconnected and roles of their synaptic aberrations in brain disorders are consistent with the cognition-relevant functions of microcircuits involving layer III neurons[18,19]. These pathophysiological changes may underlie the aberrant brain activity patterns in patients[20].

[1]Department of Neuroscience, Tufts University School of Medicine, Boston, MA, USA. [2]Department of Neurology, Yale School of Medicine, New Haven, CT, USA. [3]Department of Radiology and Biomedical Imaging, Yale School of Medicine, New Haven, CT, USA. [4]Institute of Anatomy and Molecular Neurobiology, Westfälische Wilhelms-University, Münster, Germany. [5]Department of Neuroscience, Yale School of Medicine, New Haven, CT, USA. [6]Present address: Neuroscience Department, Novartis Institutes for Biomedical Research, Cambridge, MA, USA. [7]Present address: Department of Psychology, University of Virginia, Charlottesville, VA, USA. [8]Present address: iHuman Institute, ShanghaiTech University, Shanghai, China. [9]Present address: Department of Psychiatry, University of Texas, Austin, TX, USA. [10]These authors contributed equally: Karen Perez de Arce, Adema Ribic, Dhrubajyoti Chowdhury, Katherine Watters. ✉e-mail: thomas.biederer@yale.edu

Aiming to gain insights into the molecular control of synapses in the PFC, we analyzed the adhesion molecule LRRTM1, a leucine-rich repeat transmembrane protein. This selection was guided by its disease relevance, with a haplotype upstream of *LRRTM1* over-transmitted in individuals with schizophrenia[21,22], and genetic variations in the *LRRTM1* gene associated with schizotypy[23], as well as promoter hypomethylation as an additional risk factor[24]. LRRTMs are postsynaptic proteins that promote AMPA receptor-mediated excitatory transmission and synaptic plasticity and bind pre-synaptic Neurexins[25–28]. Other trans-synaptic complexes are comprised of postsynaptic Neuroligins that also bind Neurexins, and immunoglobulin-domain proteins including SynCAM adhesion molecules, among an array of synapse organizers[3–6,10]. Recent studies have obtained the first insights into postsynaptic cooperation of adhesion molecules in developing neurons as for Neuroligins with N-Cadherin[29,30] and Neuroligin-2 with Slitrk3 at inhibitory synapses[31], as well as the presynaptic cooperation of Neurexins and protein tyrosine phosphatase σ with LRRTM4[32]. Further, LRRTMs control excitatory synapse number and transmission with Neuroligins in early but not late development[33,34].

We here aimed to understand the cooperative effects of synapse organizers when most synapses have formed and analyzed the PFC of 8–10 week old mice, when this area undergoes final maturation. Loss of LRRTM1 in knockout (KO) mice reduced excitatory synapse number in layer II/III of the mature PFC. This was accompanied by an elevation in synaptic amounts of α-Neurexins and Neuroligin-1 and the immunoglobulin superfamily member SynCAM 1, which promotes excitatory synapse number[35,36]. We focused on the increase in SynCAM 1 in the PFC which was unexpected as it forms synaptic adhesion complexes biochemically distinct from Neurexin hubs. Aiming to test the cooperation of LRRTM1 and SynCAM 1, we generated double KO (DKO) mice to abrogate the potential for redundancy. This determined that LRRTM1 and SynCAM 1 act in concert to organize synapses and dendritic spines in the medial PFC (mPFC), and their combined deletion caused a loss of synaptic Neurexin puncta in this region. Additionally, LRRTM1 and SynCAM 1 cooperated to control neuronal firing in the mPFC and synchronize brain activity. With respect to behavioral tasks involving the mPFC, the combined but not single loss of these proteins profoundly impaired sensorimotor gating, memory processes, and social interactions. Interdependent LRRTM1 and SynCAM 1 expression and concerted synaptic roles were not observed in the hippocampus. Our results show that LRRTM1 and SynCAM 1 act together to organize synapses in the PFC and brain functions. This provides insights into the cooperation of trans-synaptic complexes in the maturing brain.

## Results

### Loss of LRRTM1 reduces synapse number in the prefrontal cortex

The relevance of the PFC for cognitive functions and brain disorders[18,19] motivated us to analyze synaptic connectivity in this region. We analyzed layer II/III of the prelimbic and infralimbic PFC and the adjacent anterior cingulate in mice, which are considered homologous to the human anterior cingulate cortex[37]. These medial PFC (mPFC) areas were selected as no equivalent exists in rodents for the human dorsolateral PFC studied in disorders including schizophrenia[38]. Aiming to gain insights into roles of synapse organizing mechanisms after most synapses have formed, we tested mice at postnatal days 56–70 (P56-70) when the rodent mPFC is in the last phase of remodeling[39]. Synapse density was measured in sections from wild-type (WT) and LRRTM1 KO littermate mice immunostained for the presynaptic vesicular glutamate transporter vGlut1, a marker of excitatory cortico-cortical synapses, and the postsynaptic excitatory scaffold molecule Homer1. Puncta were identified with a machine-learning algorithm,

and sites where vGlut1 and Homer overlapped were scored as excitatory synapses (Fig. 1a). Excitatory synapse density was strongly reduced in the LRRTM1 KO mPFC by $63 \pm 6\%$ compared to WT ($p < 0.0001$) (Fig. 1b). This role of LRRTM1 in controlling synapse number in the mouse mPFC agreed with the hippocampal CA1 stratum radiatum, where LRRTM1 knockdown reduces dendritic spine density[40] and loss of LRRTM1 and LRRTM1/2 lowers excitatory synapse number[41,42].

### Interdependent expression of the synapse organizers LRRTM1 and SynCAM 1 in the PFC

We hypothesized that other synapse organizers provide partial compensation in absence of LRRTM1 to prevent further synapse loss. Presynaptic Neurexins can bind to either postsynaptic LRRTMs or Neuroligins, and we first analyzed their synaptic expression in LRRTM1 KO mice. Quantitative immunoblotting determined that α-Neurexins were strongly increased by $86 \pm 28\%$ in synaptic plasma membranes (SPMs) from LRRTM1 KO forebrain compared to WT ($p = 0.021$), and Neuroligin-1 was elevated by $43 \pm 15\%$ ($p = 0.031$) (Supplementary Fig. 1a). The adhesion molecules Neuroligin-3 and NCAM 180 were not altered upon LRRTM1 loss. Compensatory expression changes may not uniformly occur throughout the forebrain, and we next performed for a region-specific analysis. SPM fractionations require considerable tissue amounts and are not suitable for small regions like the PFC. We therefore prepared synaptosomes, a fraction comprised of pre- and post-synaptic specializations attached to each other. While synaptosomes are less pure than SPMs, they can be prepared from lower tissue amounts. Quantitative immunoblotting of synaptosomes from the PFC determined that α-Neurexins, but not β-Neurexins, were increased by $39 \pm 16\%$ ($p = 0.042$) in LRRTM1 KO mice compared to WT (Fig. 1c), similar to SPMs from total forebrain, with Neuroligin-1 showing a trend towards an increase that did not reach significance (Fig. 1d). These analyses supported that components of Neurexin complexes undergo expression changes when LRRTM1 is absent.

The immunoglobulin protein SynCAM 1 was included in this analysis of potential compensatory factors because it controls synapse number in vivo as measured in the hippocampus, is predominantly postsynaptic, and is synaptogenic similar to LRRTM1[35,36,43]. We found no evidence that SynCAM 1 is part of adhesive Neurexin complexes as it did not bind Neurexins (Supplementary Fig. 2), as reported[44]. SynCAM 1 showed a trend toward an increase in SPMs from the total forebrain of LRRTM1 KO mice compared to WT (Supplementary Fig. 1a). Analyzing the PFC, quantitative immunoblotting of synaptosomes determined that SynCAM 1 was increased in this fraction from LRRTM1 KO mice by $27 \pm 11\%$ ($p = 0.038$) compared to WT (Fig. 1e). Immunohistochemical staining in the mPFC confirmed an increase of SynCAM 1 abundance in LRRTM1 KO mice by $31 \pm 5\%$ ($p < 0.0001$) compared to WT (Fig. 1f, g). We additionally purified SPMs from the frontal cortex including the PFC, and immunoblotting confirmed an increase of synaptic SynCAM 1 in LRRTM1 KO mice by $56 \pm 16\%$ ($p = 0.017$) compared to WT (Supplementary Fig. 1b) and a strong 3.1-fold ($p = 0.028$) reciprocal increase of LRRTM1 in frontal cortex SPMs from SynCAM 1 KO mice (Supplementary Fig. 1c), using specific antibodies (Supplementary Fig. 3a).

We compared the extent to which the synaptic amounts of SynCAM 1 and LRRTM1 were increased in the hippocampus of LRRTM1 and SynCAM 1 KO mice, respectively, but found no change in SPMs from this brain region (Supplementary Fig. 4). Analysis of WT mice showed regional differences in the expression of synapse organizers, with LRRTM1 and SynCAM 1 less abundant in homogenates from the PFC than hippocampus (Supplementary Fig. 5), which may provide a higher dynamic range to facilitate increases in LRRTM1 and SynCAM 1 in PFC. These results demonstrated a PFC-selective interdependent expression of SynCAM 1 with LRRTM1.

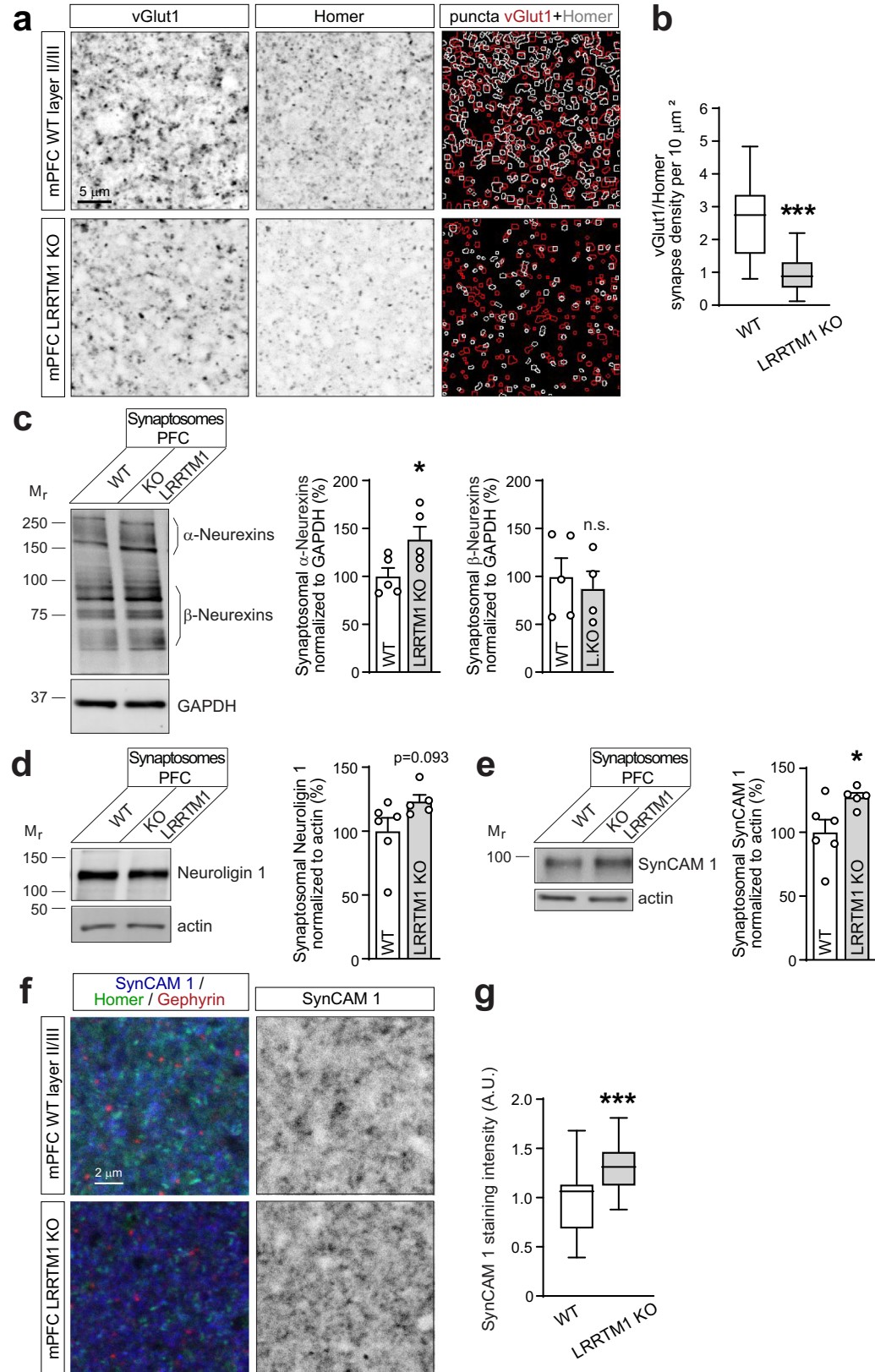

## LRRTM1 and SynCAM 1 jointly control synapse number in the prefrontal cortex

Aiming to test whether LRRTM1 and SynCAM 1 play redundant roles in organizing synapses in the maturing mPFC, we generated DKO mice to abrogate compensatory potential. Single and double KO brains showed no gross anatomical changes (Supplementary Fig. 6). We analyzed

dendritic spines as morphological correlate of excitatory synapses (Fig. 2a)[45]. Neurons of the prelimbic and infralimbic mPFC and anterior cingulate cortex were labeled with the dye DiI using biolistic delivery, and we analyzed layers II/III (Fig. 2b). As expected, most protrusions were thin spines, and their density was modestly reduced in the LRRTM1 KO mPFC by 12 ± 3.6% (p = 0.011) compared to WT littermates (Fig. 2c).

**Fig. 1 | LRRTM1 controls excitatory synapse number in the mPFC and is inter-dependently expressed with SynCAM 1. a** Representative immunostainings of mPFC layer II/III of WT (top) and LRRTM1 KO (bottom) mice at P50 for the excitatory pre- and post-synaptic markers vGlut1 and Homer1. Single optical 1.0 μm confocal sections were acquired from anterior cingulate, prelimbic, and infralimbic areas. A machine-learning algorithm traced vGlut1 and Homer puncta (red and white, respectively, in merge). Scale bar, 5 μm. **b** Fewer excitatory synapses in the mPFC of mice lacking LRRTM1. Overlapping vGlut1/Homer1 sites from images as in **a** were scored as excitatory synapses. (Student's t-test; $n = 35$ ROI WT, 64 LRRTM1 KO from 4 mice of both sexes per genotype). **c** Increase in α- but not β-Neurexins in synaptosomes from LRRTM1 KO PFC compared to WT. Left, immunoblots, 30 μg protein loaded per lane. Right, quantification normalized to loading control. (Unpaired t-test, two-tailed; $n = 5$ WT and 5 LRRTM1 KO mice for α-Neurexins; $n = 5$ WT and 4 LRRTM1 KO for β-Neurexins). **d** Left, immunoblots for Neuroligin-1 in synaptosomes from WT and LRRTM1 KO PFC. 30 μg protein loaded per lane. Right, quantification normalized to loading control. (Unpaired t-test, two-tailed; $n = 6$ WT and 5 LRRTM1 KO). **e** Increase in SynCAM 1 in synaptosomes from LRRTM1 KO PFC compared to WT. Left, immunoblots, 15 μg protein loaded per lane. Right, quantification normalized to loading control. (Unpaired t-test, two-tailed; $n = 6$ WT and 5 LRRTM1 KO). **f** mPFC sections of WT and LRRTM1 KO mice at P55 immunostained for SynCAM 1 (blue), Homer (green), and Gephyrin (red). Left, merge; right, SynCAM 1 signal. Single 1.0 μm confocal sections were acquired. **g** SynCAM 1 immunoreactivity in the mPFC layer II/III is increased in LRRTM1 KO mice compared to WT. Images as in **f** were quantified. (Unpaired t-test, two-tailed; $n = 63$ ROI from 4 WT, 66 from 4 LRRTM1 KO). *$p < 0.05$, ***$p < 0.001$, ns not significant. Error bars in bar graphs show Standard Error of the Mean. In box-and-whisker plots, the box extends from the 25th to 75th percentiles and the middle line is plotted at the median, with whiskers to maxima and minima. $M_r$ given in kilodaltons. Source data for panels **b**, **c**, **d**, **e**, and **g** are provided as a Source Data file.

SynCAM 1 KO mice showed $14 \pm 3.7\%$ fewer thin spines in this region compared to WT ($p = 0.004$) (Fig. 2c). In contrast, DKO mice exhibited a substantial reduction of thin spines, which were $45 \pm 7\%$ less frequent in DKO mPFC compared to WT. The extent of spine loss in DKO was more than additive compared to single KOs, showing a non-linear effect in DKO mPFC (One-way ANOVA, post-hoc Tukey's test, $p < 0.0001$). The density of mushroom spines was reduced in in the mPFC of single KOs to the same extent as in DKO mice (Fig. 2c), while stubby spines and filopodial protrusions did not change.

We additionally performed immunohistochemical studies of synaptic markers in layers II/III of the prelimbic area and infralimbic mPFC and the adjacent anterior cingulate cortex, and single optical sections were acquired by confocal microscopy (Fig. 2d). Machine-learning based analysis determined a reduction in postsynaptic Homer puncta density in mPFC of single LRRTM1 and single SynCAM 1 KO and DKO mice (Fig. 2e, left). This reduction in Homer puncta in the DKO mPFC was less pronounced than the loss of thin spines in DKO mice, which may be due to Homer labeling only a subset of postsynaptic specializations while DiI labels all spines. The density of vGlut1-positive presynaptic puncta was equally reduced in single LRRTM1 KO and DKO mPFC, but not changed in SynCAM 1 KO mice (Fig. 2e, right). Loss of LRRTM1 but not SynCAM 1 additionally reduced the area of Homer and vGlut1 puncta in the mPFC (Supplementary Fig. 7).

We next measured the density of excitatory synapses where vGlut1 and Homer puncta overlapped in mPFC. In agreement with the data for Homer, excitatory synapse density was lower across all KO genotypes compared to WT (Fig. 2f). This reduction was exacerbated in the mPFC of single LRRTM1 and DKO mice compared to the SynCAM 1 KO due to their loss of both vGlut1 and Homer puncta. We also asked which fraction of pre- and post-synaptic specializations are part of synapses in mPFC. Analysis of WT mice determined that $70 \pm 2\%$ of vGlut1 puncta overlapped with Homer as assessed in single optical sections, a conservative result as this analysis did not account for overlap in adjacent planes (Fig. 2g, left). Co-localization of vGlut1 with Homer was reduced in all genotypes compared to WT (Fig. 2g, left). The fraction of Homer puncta colocalized with vGlut1 was lower in DKOs than in mice lacking LRRTM1 or SynCAM 1 alone ($p < 0.0001$, LRRTM1 KO vs DKO; $p = 0.003$, SynCAM 1 KO vs DKO; ANOVA) (Fig. 2g, right), even though vGlut1 number was equally reduced in single LRRTM1 KO and double KO mice. This provided evidence for a concerted role of both synapse organizers in aligning post- with pre-synaptic excitatory sites.

To analyze inhibitory synapses in the mPFC, we quantified immunohistochemical stainings for presynaptic vGAT and postsynaptic Gephyrin. This determined across KO genotypes a lower density of sites where vGAT and Gephyrin colocalize compared to WT, and smaller Gephyrin puncta (Supplementary Fig. 8). This reduction may reflect a homeostatic response to excitatory synapse loss and not a direct role in inhibitory synapse organization as both LRRTM1 and SynCAM 1 are part of the excitatory synaptic cleft[7,25,43].

Our results raised the question to what extent excitatory synaptic specializations were impacted in other brain regions. We therefore extended our analysis of spine density changes to the hippocampal CA1 stratum radiatum (Fig. 2h, i), where LRRTM1 is preferentially expressed[40,46]. Here, thin and mushroom spine densities were equally reduced in single LRRTM1 and SynCAM 1 KO mice, and their combined deletion did not further reduce either spine type density in CA1 unlike the more-than-additive loss of thin spine density in LRRTM1 and SynCAM 1 DKO mPFC. These results supported concerted roles of LRRTM1 and SynCAM 1 in organizing excitatory synapses in the mPFC but not other forebrain regions.

## A cooperative mechanism involving LRRTM1 and SynCAM 1 modulates Neurexin distribution

The impact of combined loss of LRRTM1 and SynCAM 1 on thin spine density together with synaptic alignment raised the question whether presynaptic organizers are impacted in DKO mPFC. LRRTM proteins bind Neurexins, which serve as presynaptic hubs specifying synaptic properties[47]. Neurexin 1 is present at about 40% of mature excitatory synapses[48,49]. The mechanisms of Neurexin sorting to synaptic sites are being defined[50–52], but how the synaptic distribution of Neurexins is controlled at mature synapses remains incompletely understood. Using specific antibodies (Supplementary Fig. 3b, c), we performed immunostainings to quantify Neurexin staining in the mPFC layer II/III in WT mice. Neurexins exhibited punctate labeling that partially colocalized with the presynaptic active zone marker Bassoon (Fig. 3a) and vGlut1 (Fig. 3b). Analysis of mPFC areas in LRRTM1 KO mice determined that the total density of Neurexin puncta was reduced in DKO mice by $20 \pm 6.6\%$ compared to WT (ANOVA, $p = 0.006$) (Fig. 3c).

We also analyzed the excitatory synaptic population of Neurexins by co-immunolabeling with vGlut1. In WT mPFC, $31 \pm 2\%$ of Neurexin puncta colocalized with this marker (Fig. 3d). This synaptic Neurexin population was increased by $45 \pm 11\%$ in LRRTM1 KO mice compared to WT (ANOVA, $p = 0.0002$) (Fig. 3d), consistent with the increase in α-Neurexin in synaptic fractions from LRRTM1 KO mice (Fig. 1c and Supplementary Fig. 1a). This increase in the fraction of synaptic but not total Neurexin puncta in the mPFC of LRRTM1 KO mice may be due to an upregulation of synapse organizers that control the synaptic abundance of Neurexins. Combined deletion of LRRTM1 and SynCAM 1 caused a loss of the synaptic Neurexin population by $28 \pm 3\%$ compared to single SynCAM 1 KO mice (ANOVA, $p = 0.019$) and a trend towards a loss compared to WT, similar to the loss of total Neurexin puncta in DKO mice (Fig. 3c, d). This combined requirement of LRRTM1 and SynCAM 1 for physiological Neurexin puncta densities provided additional evidence that they cooperate in organizing presynaptic sites.

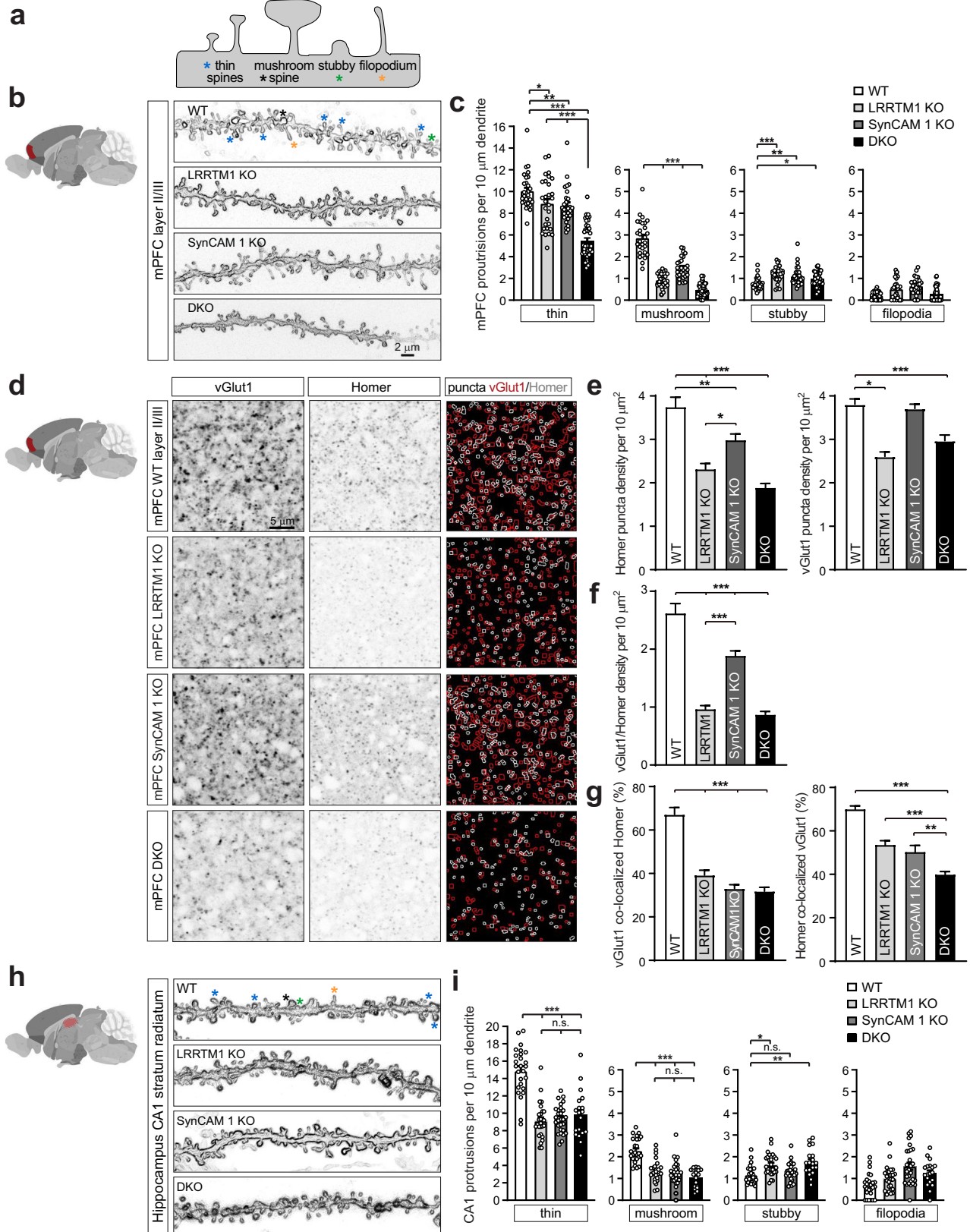

### Neuronal firing in the mPFC is altered in absence of LRRTM1 and SynCAM 1

Our next question was whether activity patterns were altered in LRRTM1/SynCAM 1 DKO mice. We first addressed effects of LRRTM1 and SynCAM 1 loss on neuronal activity by performing extracellular recordings of multi-unit activity (MUA) from layer II/III of both prelimbic and infralimbic areas from awake, head-fixed mice (Fig. 4a). These recordings represent the aggregate signal of local spiking activity, including principal cells and interneurons. MUA was elevated in LRRTM1 KO mice compared to WT littermates ($p < 0.0001$; WT, $146 \pm 4.1$ spikes/s; LRRTM1 KO, $171 \pm 4.2$ spikes/sec) (Fig. 4b). SynCAM 1 KO mice showed no detectable difference in spontaneous MUA

**Fig. 2 | Spine loss and excitatory synapse aberrations in mPFC in absence of LRRTM1 and SynCAM 1. a** Diagram of dendritic spine classification. **b** Left, diagram marking analyzed PFC in red. Right, DiI-labeled secondary and tertiary dendrites in mPFC layer II/III of WT, LRRTM1 KO, SynCAM 1 KO, and DKO mice. Asterisks, color code as in **a**. Scale bar, 2 μm. **c** Fewer thin and mushroom spines in mPFC of LRRTM1 KO and SynCAM 1 KO mice compared to WT quantified from images as in **b**. DKO mice exhibited a larger, more than additive decrease in thin spines. (One-way ANOVA with post-hoc Tukey's test; $n = 29$ dendritic segments from 3 WT, 29 from 3 LRRTM1 KO, 28 from 3 SynCAM 1 KO, 41 from 3 DKO). **d** Left, diagram marking analyzed mPFC. Right, representative immunostainings for excitatory pre- and post-synaptic markers vGlut1 and Homer1 in mPFC layer II/III (red and white, respectively, in algorithm-traced merge). Single optical 1.0 μm confocal sections were obtained. Scale bar, 5 μm. **e** Lower densities of excitatory postsynaptic Homer (left) and presynaptic vGlut1 (right) in single LRRTM 1 and LRRTM1/SynCAM 1 DKO mPFC compared to WT. Fewer Homer but no change in vGlut1 sites was observed in single SynCAM 1 KO mPFC. Images as in **d** were quantified. (Homer, WT vs. LRRTM1 KO, $p < 0.0001$; WT vs SynCAM 1 KO, $p = 0.006$; WT vs. DKO, $p < 0.0001$; LRRTM1 KO vs SynCAM 1 KO, $p = 0.022$; vGlut1, WT vs. LRRTM1 KO, $p < 0.0001$; WT vs. DKO, $p < 0.0001$) (One-way ANOVA with post-hoc Tukey's test; $n = 35$ ROI from 4 WT, 64 from 4 LRRTM1 KO, 35 from 4 SynCAM 1 KO, and 64 from 4 DKO). **f** The density of excitatory synapses, defined as sites where vGlut1 and Homer puncta co-localize, is reduced in single and double KO mice. Images as in **d** were quantified. WT control and LRRTM1 KO data replicate the results in Fig. 1b. (One-way ANOVA with post-hoc Tukey's test; WT vs. all, $p < 0.0001$) Number of analyzed ROI and mice as in **e**. **g** Single or combined loss of LRRTM1 and SynCAM 1 reduces the extent to which vGlut1 puncta co-localize with Homer (left) and the co-localization of Homer with vGlut1 (right). DKO mice exhibit the strongest reduction of Homer co-localization with vGlut1. Images as in **d** were quantified. (One-way ANOVA with post-hoc Tukey's test; vGlut 1 with Homer, WT vs. all, $p < 0.0001$; Homer with vGlut1, SynCAM 1 KO vs DKO, $p < 0.0001$; WT vs. all others, $p < 0.0001$) Number of analyzed ROI and mice as in **e**. **h** Dendritic spine density in the hippocampal CA1 stratum radiatum. Left, diagram with hippocampus marked in red. Right, representative images of DiI-labeled dendritic shafts and spines in CA1 of WT, LRRTM1 KO, SynCAM1 KO, and DKO mice. Asterisks, spine color code as in **a**. Scale bar, 2 μm. **i** Spine density losses in CA1 stratum radiatum of DKO mice was indistinguishable from single KOs. Images as in **h** were quantified. (One-way ANOVA with post-hoc Tukey's test; $n = 26$ dendritic segments from 3 WT mice, 26 from 3 LRRTM1 KO, 28 from 3 SynCAM 1 KO, 19 from 3 DKO). $*p < 0.05$, $**p < 0.01$, $***p < 0.001$, n.s., not significant. Error bars in bar graphs show Standard Error of the Mean. Source data for panels **c**, **e**–**g** and **i** are provided as a Source Data file.

(SynCAM 1 KO, $143 \pm 4.8$ spikes/s). In contrast, double LRRTM1/SynCAM1 KO mice exhibited significantly lower MUA ($p < 0.0001$; DKO, $108 \pm 3.4$ spikes/s).

We extended this analysis to KO effects on principal cells and inhibitory interneurons. Single unit responses were isolated from the MUA recordings and sorted into broad and narrow spikes associated with the firing of principal cells and inhibitory interneurons, respectively, as described[53,54]. This analysis showed a decrease in the firing rate of broad-spiking units in DKO mice compared to WT as well as single LRRTM1 and single SynCAM 1 KO mice (Fig. 4c). These data supported lowered firing of putative excitatory neurons in the DKO PFC and agreed with the MUA results and the reduction in excitatory synapses and dendritic spines in these mice. Single LRRTM1 and SynCAM 1 KO mice exhibited trends toward lower firing rate of broad spiking units that were not significant. We did not isolate sufficient numbers of narrow spiking units due to their low abundance, and instead analyzed the staining intensity for Parvalbumin (PV) in PV-positive interneurons which is a correlate of their activity[55]. Our immunohistochemical analysis of layer II/III of prelimbic and infralimbic areas determined a substantial increase in PV intensity in the LRRTM1 KO (Fig. 4d). SynCAM 1 was required for this phenotype as PV staining in DKO mice, albeit slightly higher than in WT, was significantly lower when compared to LRRTM1 KO mice. These results supported alterations in PV network activity in absence of LRRTM1 and agreed with the altered MUA firing rates in LRRTM1 KO and DKO mice.

## Altered synchronization of brain activity fluctuations in DKO mice

Spontaneous changes in neuronal activity can be assessed by fMRI of resting state fluctuations in the spontaneous blood oxygenation level-dependent (BOLD) global signal (GS). A high amplitude of GS fluctuation in consistent with greater uncoupling of blood flow and metabolism and points to lower neuronal activity[56]. GS fluctuations are coupled to behavioral states and neural activity in the cerebral cortex of primates[57] and schizophrenia patients exhibit higher GS power, indicative of altered connection synchronization[58]. fMRI determined that GS amplitude was elevated in mouse brains when SynCAM 1 was deleted (Fig. 5a, b). The single loss of LRRTM1 did not change GS ($p = 0.0003$ SynCAM 1 KO, $p = 0.091$ LRRTM1 KO, $p = 0.24$ interaction), but the greatest GS amplitude was measured in DKO mice (post-hoc t-test, two-sample, one-tailed, equal variance; $p = 0.014$ DKO vs SynCAM 1 KO) (Fig. 5a, b). LRRTM1 hence contributed to the control of GS with SynCAM 1, a redundant role only apparent under the sensitized condition of SynCAM 1 loss. These data showed that SynCAM 1 and LRRTM1 together control brain activity fluctuations and restrict the mass synchronization reflected in GS.

## Combined loss of SynCAM 1 and LRRTM1 KO impacts social interactions

The functions of LRRTM1 and SynCAM 1 in controlling neuronal connectivity and activity levels could impact behavior. WT, single KO, and DKO mice were initially tested for locomotor activity, which was unaltered in the open field (Supplementary Fig. 9a). No changes were observed in motor coordination and motor skill learning using the Rotarod test (Supplementary Fig. 9b). Exploratory behavior was assessed by measuring rearing, when the animal is moving in an open arena and entering search phases[59]. This behavior is sensitive to stress and anxiety and was not altered in any genotype (Supplementary Fig. 9c).

Anxiety was further assessed using the elevated plus maze (Fig. 6a). Single or double KO mice showed no difference to WT in the number of entries into the open arms versus closed arm entries (Fig. 6b). While a previous study of LRRTM1 KO mice reported a phenotype[60], this may be due to using mice up to twice the age of animals tested here and different genetic backgrounds, which can impact behaviors including after loss of synapse organizers[61]. The absence of gross behavioral changes enabled the analysis of higher brain functions.

We next analyzed whether concerted functions of LRRTM1 and SynCAM 1 manifest in DKO mice as altered social interactions. We chose this paradigm because it involves activation of neurons in the mPFC[62] and the fact that impaired social interactions are a diagnostic criterion for schizophrenia, making them relevant for assessing disease-linked roles of LRRTM1. We used a 3-chamber test in which a mouse chooses between exploring a chamber that holds an object or a chamber with an unfamiliar mouse (Fig. 6c). Single KO mice lacking LRRTM1 or SynCAM 1 performed indistinguishably from WT mice and engaged the stranger for longer than the object (Fig. 6d). In contrast, DKO mice did not prefer the stranger mouse over an object. SynCAM 1 and LRRTM1 hence together modulate aspects of social interactions.

## Aberrant cognition-relevant behaviors in absence of LRRTM1 and SynCAM 1

We probed cognitive functions by testing memory processes and attention. These tasks involve the mPFC, with prefrontal and anterior cingulate cortex engaging the hippocampus for consolidating memory, including of spatial reference memory[63,64]. We investigated memory processes using the Morris water maze. Mice were trained in a water-filled tank to locate a quadrant that contained a hidden platform,

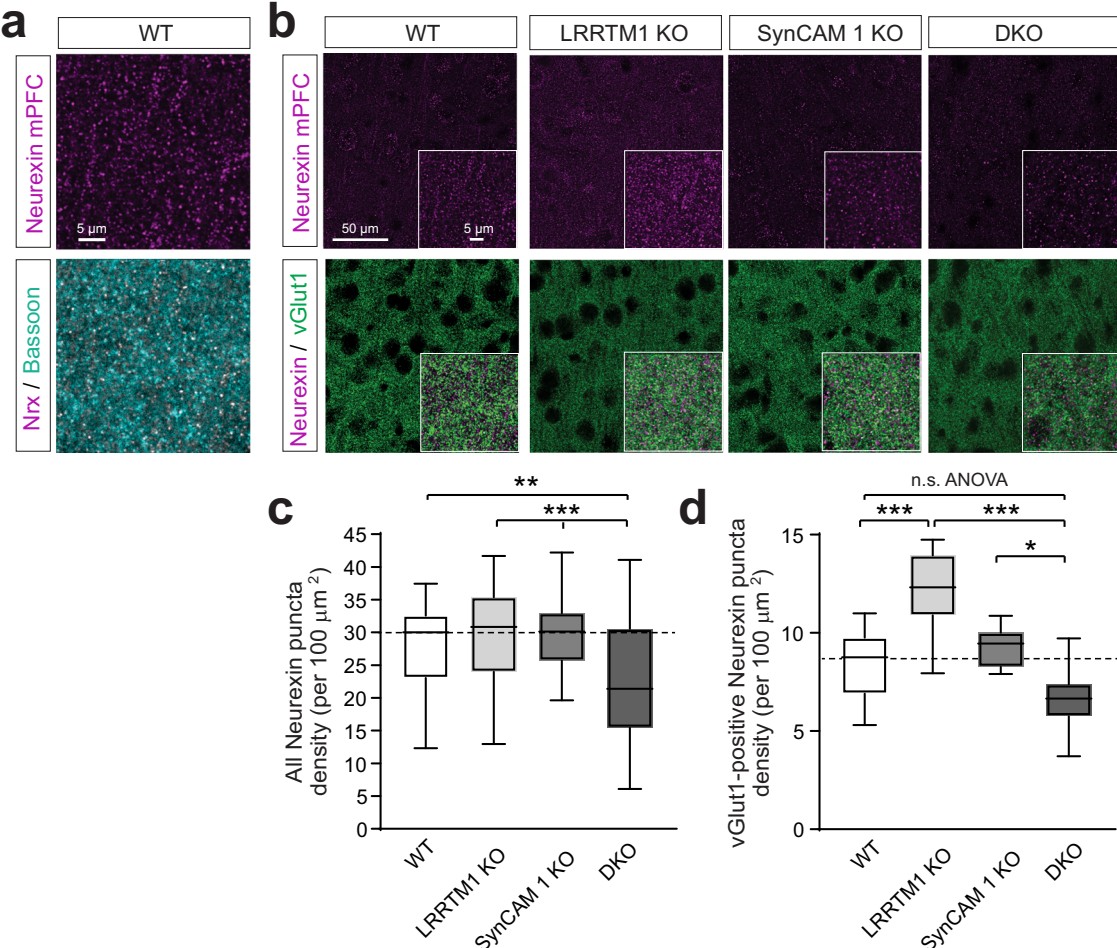

**Fig. 3 | LRRTM1 and SynCAM 1 engage in the trans-synaptic control of Neurexin expression. a** Immunostainings of mPFC layer II/III in P55 WT mice for Neurexins (magenta) and the active zone marker Bassoon (green). White marks colocalized sites in the merge. Single 1.0 μm confocal sections were obtained. The shown example is from prelimbic PFC and representative of results obtained in three independent experiments. Scale bar, 5 μm. **b** Top row, Neurexin immunostainings from WT, LRRTM1 KO, SynCAM 1 KO and DKO mice in PFC layer II/III in anterior cingulate, prelimbic, and infralimbic areas at P50. Bottom row, co-staining Neurexin (magenta) with vGlut1 (green). Single 1.0 μm confocal sections were acquired. Insets show enlarged areas. **c** Neurexin puncta loss in DKO mPFC. Total Neurexin puncta number was quantified in anterior cingulate, prelimbic and infralimbic areas from images as in **b**. (One-way ANOVA with post-hoc Tukey's test; WT vs. DKO, $p = 0.006$) ($n = 38, 40, 42, 43$ ROI from each 4 WT, LRRTM1 KO, SynCAM 1 KO, DKO). **d** The number of presynaptic excitatory Neurexin puncta positive for vGlut1 is increased in LRRTM1 KO mPFC and combined KO of LRRTM1 and SynCAM 1 causes a loss of excitatory Neurexin puncta compared to single KO mice. (One-way ANOVA with post-hoc Tukey's test; WT vs. SynCAM 1 KO, $p = 0.019$; $n$ as in **c**). *$p < 0.05$, **$p < 0.01$, ***$p < 0.001$. In the plots, the box extends from the 25th to 75th percentiles and the middle line is plotted at the median, with whiskers to maxima and minima. Source data for panels **c** and **d** are provided as a Source Data file.

learning visual cues to navigate (Fig. 7a). The latency to reach a visible platform was not different between mice of the tested genotypes (Supplementary Fig. 9d). LRRTM1 and SynCAM 1 single and double KO mice had indistinguishable swim speeds and attained over 5 training days the same time as WT animals to reach the hidden platform (Fig. 7b). Motivation to reach the target and memory acquisition were hence unaffected by the loss of LRRTM1 and SynCAM 1 in this test. One day after training was completed, the platform was removed, and mice were tracked while navigating to the escape quadrant where the platform had been hidden. WT and single KO mice achieved this task (Fig. 7c), and we did not observe an impairment reported upon LRRTM1 loss in a different genetic background[41]. The most pronounced choice of trained WT mice is to prefer the target quadrant over the opposite quadrant. In contrast to the single KO genotypes, DKO mice did not show this preference (Fig. 7c, SW vs. NE quadrant). This supported that combined but not single loss of LRRTM1 and SynCAM 1 impairs spatial memory recall.

Finally, we probed attentive processing and analyzed prepulse inhibition (PPI). This paradigm tests sensorimotor gating by measuring whether a startle response induced by a loud acoustic stimulus is lowered by a preceding, less loud stimulus (Fig. 7d). This behavior was chosen because it involves PFC circuits[65,66]. Further, it relates to the diagnostic criterion of cognitive dysfunction including attention deficits in schizophrenia patients, whose startle response remains unaltered by a preceding acoustic stimulus[67]. A low-intensity acoustic prepulse stimulus inhibited in WT mice the startle response caused by a subsequent loud stimulus, as expected (Fig. 7e). Following acoustic prepulses of increasing intensities, WT mice showed progressive inhibition of the startle response, as expected (Fig. 7e). Normal startle responses were observed in single KO mice lacking LRRTM1 or SynCAM 1, with exception for a small deficit in SynCAM 1 KO mice at the lowest acoustic prepulse. We did not observe PPI impairments previously reported after loss of LRRTM1 in the thalamus[68], which may be due to differences in the age or compensation in other brain regions. Notably, DKO mice exhibited diminished PPI across all prepulse intensities, even at the loudest prepulse (Fig. 7e). These data support that LRRTM1 and SynCAM 1 act in concert to support attentive processing. The combined loss of these synapse organizers therefore unmasked impairments across cognition-relevant behavioral domains.

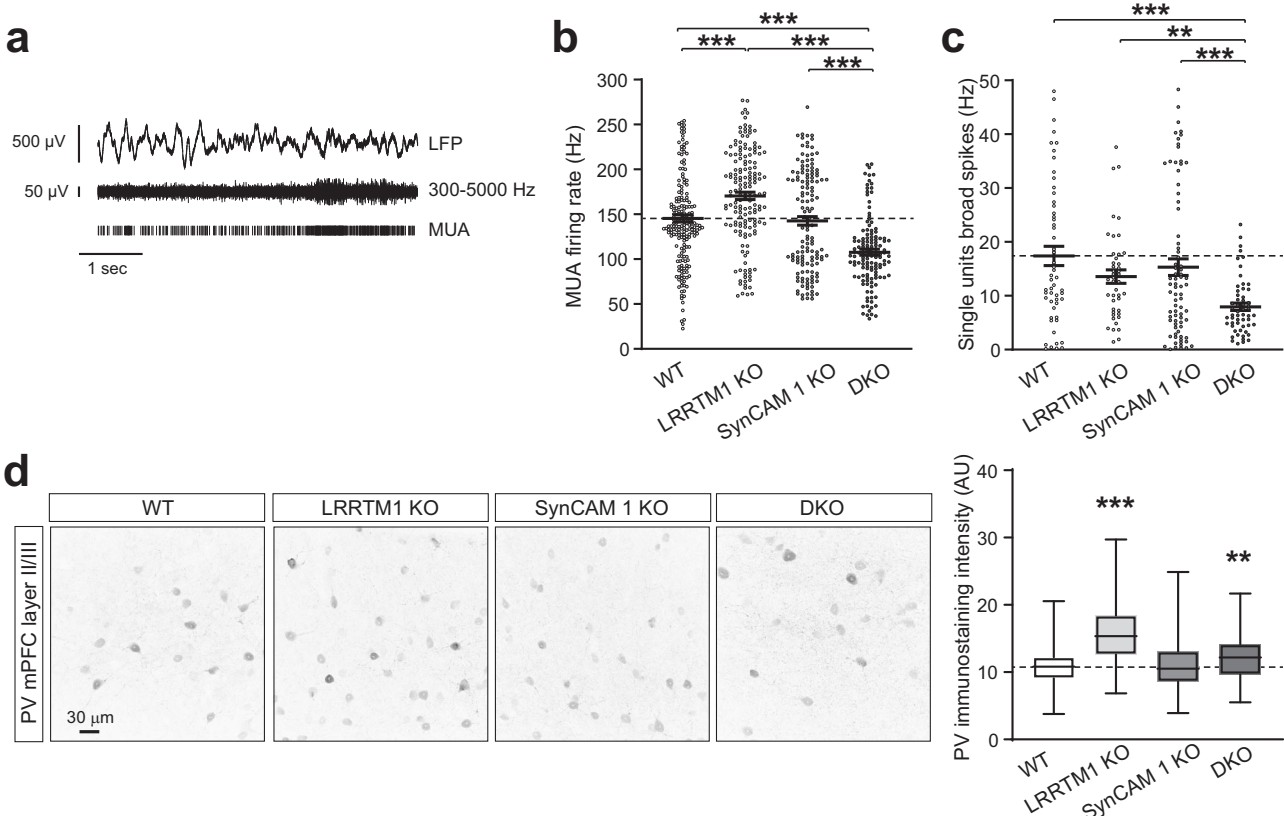

**Fig. 4 | Aberrant neuronal activity in the mPFC of mice lacking LRRTM1 and SynCAM 1. a** Multi-unit activity (MUA, middle and bottom traces) was extracted from band-pass filtered local field potentials (LFPs, top trace) recorded from layer II/III of the mPFC of awake mice. MUA recordings encompass firing of excitatory and inhibitory neurons. **b** Single LRRTM1 loss elevates MUA firing rate and combined loss of LRRTM1 and SynCAM 1 reduces it. Firing rate was calculated as average spiking rate over time (in sec) recorded from mice around P50 as in **a**. (Kruskal-Wallis ANOVA with Dunn's post hoc test; $n = 199$ traces from 5 WT mice, 160 traces from 4 LRRTM1 KO, 137 traces from 3 SynCAM 1 KO, and 132 traces from 3 DKO). **c** Analysis of single unit responses shows fewer broad spikes associated with the firing of principal cells in mice lacking both LRRTM1 and SynCAM 1, but not in single KO mice. (One-way ANOVA with Dun's post hoc test; $n = 54$ units from WT,

44 from LRRTM1 KO, 78 from SynCAM 1 KO, and 56 from DKO). **d** Parvalbumin (PV) staining in fast-spiking PV-positive interneurons is strongly increased in LRRTM1 KO and modestly in DKO. Left, representative immunostainings for PV in mPFC layer II/III. Images show background subtracted maximum intensity projections of z-stacks. Right, PV intensity quantification from images as on the left. (One-way ANOVA with Dunn's post hoc test; images from 6 WT mice, 3 LRRTM1 KO, 4 SynCAM 1 KO, 3 DKO). Asterisks mark difference to WT. $**p < 0.01$, $***p < 0.001$. Error bars in **b** and **c** show Standard Error of the Mean. In the plots in **d**, the box extends from the 25th to 75th percentiles and the middle line is plotted at the median, with whiskers to maxima and minima. Source data for panels **b**–**d** are provided as a Source Data file.

## Discussion

This study of trans-synaptic cooperativity obtained five key findings. First, the synaptic adhesion molecules LRRTM1 and SynCAM 1 are interdependently expressed in the PFC. Second, removing both LRRTM1 and SynCAM 1 reduces spine density and the alignment of excitatory post- with pre-synaptic markers beyond the sum of aberrations in single KO mice. Third, Neurexins expression is jointly modulated by these molecules. Fourth, LRRTM1 and SynCAM 1-dependent mechanisms cooperate to promote neuronal activity. Fifth, this cooperation manifests on a behavioral level and combined but not single loss of these molecules impairs memory, attention, and social interactions. These results demonstrate that synapse-organizing mechanisms involving LRRTM1 and SynCAM 1 act in concert across synaptic, activity, and behavioral domains.

Impairments of excitatory synapses contribute to psychiatric disorders as supported by the convergence of genome-wide association studies, de novo mutations, and dysregulated gene expression on synaptic risk factors[11,69]. Studies of neurodevelopmental disorders further highlight the disease relevance of synapse organizers[47,70]. We here addressed how trans-synaptic interactions contribute to synapse impairments. We investigated this in the rodent mPFC that shares physiological and cognition-relevant functions with human dorsolateral

PFC[71] and analyzed mice at 8–10 weeks, when their mPFC undergoes late anatomical and functional maturation[39]. We focused on layer II/III due to its lower dendritic spine number and circuit aberrations in schizophrenia[16–18]. This analysis determined that LRRTM1 is required for physiological synapse density in layer II/III of the mature mPFC. We also assessed other synapse organizers and found that synaptic α-Neurexins and Neuroligin-1 are increased LRRTM1 KO forebrain, indicating compensatory responses across Neurexin hubs. Further, our experiments measured increased synaptic SynCAM 1 in LRRTM1 KO PFC.

This unexpected interdependency of the expression of synaptic cleft complexes in the mPFC motivated us to address whether the SynCAM 1 increase in LRRTM1 KO mPFC synapses can be a compensatory response. We tested this in LRRTM1 and SynCAM 1 DKO mice to abrogate redundancy. Indeed, DKO mice exhibited a pronounced spine loss in the mPFC layer II/III that exceeded the sum of the comparatively subtle reductions in single KOs. We also determined using immunohistochemistry that LRRTM1 and SynCAM 1 are individually required for normal numbers of excitatory postsynaptic sites and together control the extent to which these sites co-localize with presynaptic terminals. While a majority of SynCAM 1 is postsynaptic, at least in the hippocampal CA1 region[43], it can be considered that both its post- and presynaptic populations are involved. SynCAM 1 does not bind LRRTMs[72]

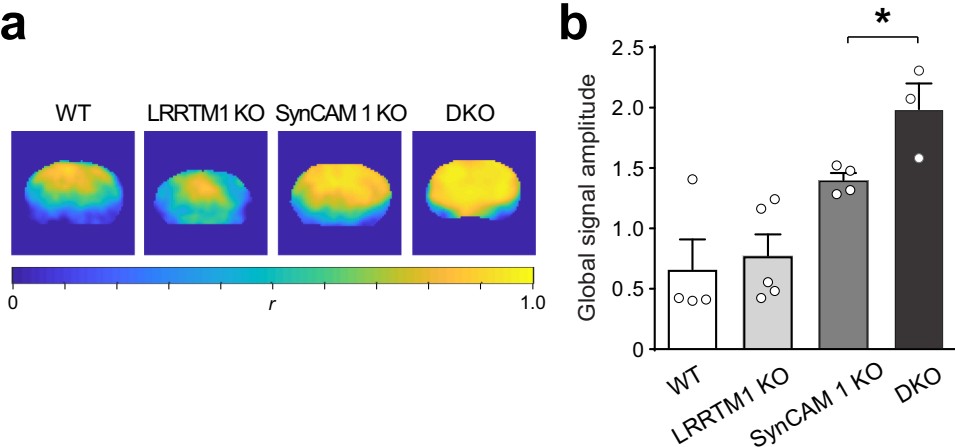

**Fig. 5 | Altered synchronization of brain activity in LRRTM1 and SynCAM 1 DKO mice. a** Maps of the Pearson's correlation coefficient *r* between the fMRI global signal (GS) and all voxels in the whole brains of representative mice from each group. One coronal slice, ~−4 mm Bregma, is shown per mouse. Correlation with GS is stronger in SynCAM 1 KO and DKO groups. **b** SynCAM 1 and LRRTM1 together modulate spontaneous, synchronous brain activity patterns as shown by measuring

GS from images as in **a**. GS amplitude was calculated as the standard deviation of the mean signal from the entire brain. DKO mice have the highest increase in GS amplitude, indicating more uniform synchronization. Source data are provided as a Source Data file. Error bars show Standard Error of the Mean. (N-way ANOVA with Tukey's post hoc test, *p* = 0.0003 SynCAM 1, *p* = 0.091 LRRTM1, *p* = 0.24 interaction; *n* = 4 WT, *n* = 5 LRRTM1 KO, *n* = 4 SynCAM 1 KO, *n* = 3 DKO).

or Neurexins[44] (and this study), and we presume that these concerted effects occur via their distinct trans-synaptic complexes. Our results support a cooperative potential of LRRTMs in the maturing brain that expands their functional relevance after they organized in early development excitatory synapses with Neuroligins[33,34,40]. These findings also provide insight into the requirement of LRRTM1 in the mature brain, relevant due to the differential roles of LRRTMs in developing and maturing neurons[42]. Further, the fact that combined deletion of LRRTM1 and SynCAM 1 substantially impacts the mPFC but not the hippocampus indicates that the mPFC may have less compensatory potential than other brain regions. These findings support the need for region-select analyses of synapse organizers.

How the absence of SynCAM 1 or LRRTM1 results in the other molecule's increase at mPFC synapses remains to be determined. Proximity labeling supports that SynCAM 1 is in the vicinity of LRRTM1 at the same synapses[7] and LRRTM1 antibodies suitable for immunostaining will enable testing whether their cooperation involves the same or distinct synaptic populations. A slot model in which the number of both adhesion molecules in postsynaptic membranes is set by binding site availability can be considered, but is difficult to reconcile with the distinct subsynaptic localizations of LRRTMs in confined, central postsynaptic domains while SynCAM 1 occupies a zone at the synaptic edge[43,46,73]. Alternatively, functional impairments upon loss of one of these proteins could be restored to a setpoint by homeostatic elevation of the other.

The increase in the density of synaptic, vGlut1-positive Neurexin puncta in the mPFC of LRRTM1 KO mice was consistent with their higher synaptic α-Neurexin amount. Interestingly, this phenotype required SynCAM 1 as its additional loss abrogated the increase of synaptic Neurexin puncta number seen in absence of LRRTM1. The underlying mechanism may involve presynaptic scaffold and signaling proteins that bind both trans-synaptic SynCAM/SynCAM and Neurexin complexes[36,74]. It can be considered that the higher abundance of SynCAM complexes in LRRTM1 KO mPFC indirectly enhances the recruitment of presynaptic Neurexins, which in turn would promote the number of their puncta. DKO mice would lack such a mechanism.

On a physiological level, our recordings from layer II/III in the prelimbic and infralimbic mPFC provide evidence that these two synapse organizers control the activity of neurons in the mPFC. Loss of both LRRTM1 and SynCAM 1 substantially reduced neuronal firing and excitatory single unit events, in agreement with the reduction in

dendritic spines in DKO mice. The LRRTM1 KO-specific increase in MUA that represents all local excitatory and inhibitory spiking activity could be due to a higher activity of Parvalbumin-positive interneurons that was indicated by increased Parvalbumin staining. This phenotype required SynCAM 1 as it is absent on DKO mice, and loss of SynCAM 1-dependent excitatory inputs to PV-positive interneurons[75] could contribute to silencing these cells in DKO mice. fMRI data further supported altered activity patterns in DKO mice.

In agreement with impaired synaptic connectivity, multiple behavioral functions were impacted in DKOs with no apparent deficits in single KO mice. Among these behaviors was attentive processing, which is controlled by parieto-frontal networks. In addition, our spatial memory analysis supports that SynCAM 1 functions with LRRTM1 to promote memory consolidation or recall, which was proposed to involve the mPFC[63]. The DKO impairments are reminiscent of schizophrenia patients, including deficits in attention, the ability to form spatial memories, and lower IQ[67,76,77]. Our DKO results also agree with the role of balanced connectivity in the mPFC in social interactions in humans and rodents[78]. The finding that the synapse loss in LRRTM1 KO mice did not cause apparent impairments may reflect that this loss remains above a minimum threshold, which is crossed by the additional synapse reduction in DKO mice. Synapses in DKO mice may also be less functional than in WT because of the inability to upregulate SynCAM 1, which can now be tested. These behavioral phenotypes in DKO mice could hence result from impaired connectivity or excitation/inhibition balance deficits.

Together, our results reveal the relationship of the adhesion molecules LRRTM1 and SynCAM 1 in organizing synapses in the mPFC. This shows that molecularly distinct synaptic cleft complexes can act together to control neuronal connectivity, providing for a cooperation that could profoundly expand their individual functions. In addition, this study provides evidence that synapse organizers can have concerted functions in select regions. Our results point to a particular vulnerability of the PFC to an imbalance of synapse organizing pathways, in agreement with the relevance of cortical processing networks for brain disorders[79].

## Methods
### Animal procedures
All animal procedures undertaken in this study were approved by Institutional Animal Care and Use Committees (Tufts University,

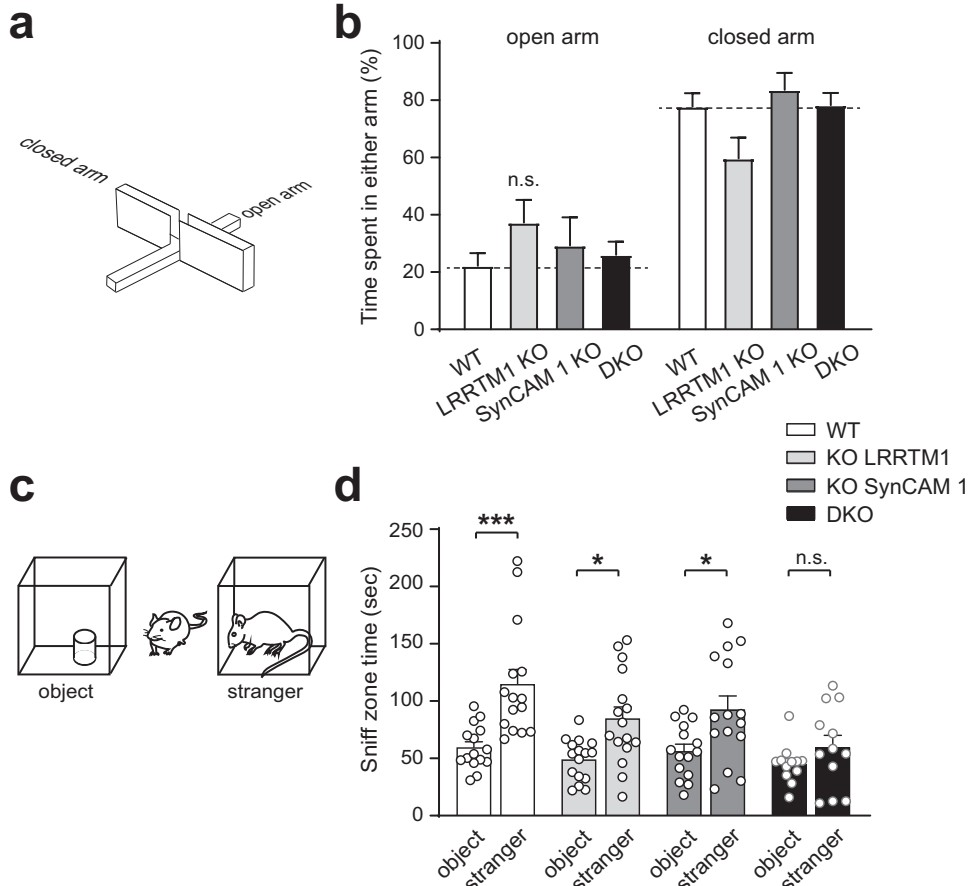

**Fig. 6 | No evidence for changes in anxiety but impaired social interactions upon combined deletion of LRRTM1 and SynCAM 1. a** Model of the elevated plus maze to assess anxiety. Mice are placed in the center of a plus maze and number of entries and length of time spent in open versus closed arms were scored. **b** WT mice prefer to spend time in the closed arm of the elevated plus maze, and single LRRTM1 and SynCAM 1 KO and DKO mice behave like WT. Dotted lines mark WT times. (One-way ANOVA with Tukey's post hoc test; $n = 16$ WT, 18 LRRTM1 KO, 12 SynCAM 1 KO, 11 DKO mice). **c** Model of the social preference test. Mice are placed between an object and a stranger mouse and sniff zone time around each is recorded. **d** While LRRTM1 KO and SynCAM 1 KO mice behave like WT and spend more time exploring a stranger mouse than an object, DKO mice show no significant preference. (One-way ANOVA with Tukey's post hoc test; $n = 15$ WT, 16 LRRTM1 KO, 15 SynCAM 1 KO, 12 DKO mice). *$p < 0.05$, ***$p < 0.001$, n.s., not significant. Error bars show Standard Error of the Mean. Source data for panels **b** and **d** are provided as a Source Data file.

Boston, Massachusetts, USA and Yale University, New Haven, Connecticut, USA) in compliance with NIH guidelines and the Landesamt für Natur, Umwelt und Verbraucherschutz (North Rhine-Westphalia, Germany). Male and female mice were analyzed in this study except for behavioral experiments, where only male mice were tested. Mice were housed in an environment of between 40 and 60% relative humidity and an ambient temperature of approximately 21 °C/70 °F.

### Antibodies

Immunoblotting was performed with antibodies against LRRTM1 (R&D Systems Cat# AF4897, RRID:AB_10643427; 1:800), SynCAM 1 (MBL Laboratories Cat# CM004-3, clone 3E1, RRID:AB_592783; 1:1500), Neurexins (Millipore Cat# ABN161, RRID:AB_10917110; 1:300), Neuroligin-1 (Synaptic Systems Cat#129 111, clone 4C12, RRID:AB_887747; 1:500), Neuroligin-3 (NeuroMab clone N110/29, RRID:MMRRC_066080-UCD; 1:500), NCAM 180 (Sigma Cat# C9672, RRID:AB_1079450; 1:500), GAPDH (Millipore Cat# MAB374, clone 6C5, RRID:AB_2107445; 1:5000), and actin (MP Biomedicals Cat# 0869100, clone C4, RRID:AB_2335127; 1:5000).

Antibodies for immunohistochemistry detected vGlut1 (Neuro-Mab clone N28/9, RRID:MMRRC_065995-UCD; 1:800), Homer 1/2/3 (Synaptic Systems Cat# 160 003, RRID:MMRRC_065995-UCD; 1:500), Neurexins (Millipore Cat# ABN161, RRID:AB_10917110; 1:200), SynCAM 1 (MBL Laboratories Cat# CM004-3, clone 3E1, RRID:AB_592783;

1:1000), PSD-95 (Cell Signaling Cat# 3409, RRID:AB_1264242; 1:500), Parvalbumin (Swant, Cat# PVG-213, RRID:AB_2650496; 1:500) and MAP2 (Millipore Cat# MAB3418, RRID:AB_94856; 1:3000).

Primary antibodies for immunocytochemistry were directed against Synapsin 1 (Synaptic Systems Cat# 106 001, RRID:AB_887805; 1:1000) and Neurexin (Millipore Cat# ABN161, RRID:AB_10917110; 1:1000). Secondary antibodies were conjugated with AlexaFluor dyes 488, 555, or 647 (Thermo Fisher), using Ig-subtype specific antibodies to detect monoclonal antibodies, and for immunocytochemistry CyTM3-conjugated goat-anti-rabbit IgG (Jackson ImmunoResearch Cat# 111-165-003, RRID:AB_2338000).

### Biochemical procedures

Biochemical analyses were performed in mice at P56-63. mPFC samples were collected from the most caudal level of the rodent PFC, just anterior to bregma as marked in Fig. 4 of Laubach et al.[37]. Tissue samples were flash-frozen in liquid $N_2$ and rapidly homogenized using microtip-aided sonication in Hepes pH 7.4 (50 mM), urea (8.0 M), and PMSF (0.5 mM).

Synaptic plasma membranes were purified by the method of Jones and Matus[80], with modifications[36]. Synaptosomes were prepared from prefrontal cortices or hippocampi of individual animals. Tissue was homogenized on ice in ice-cold suspension buffer (320 mM sucrose, 10 mM HEPES, 1 mM PMSF, 1.5 μM Pepstatin A, 2.1 μM Leupeptin,

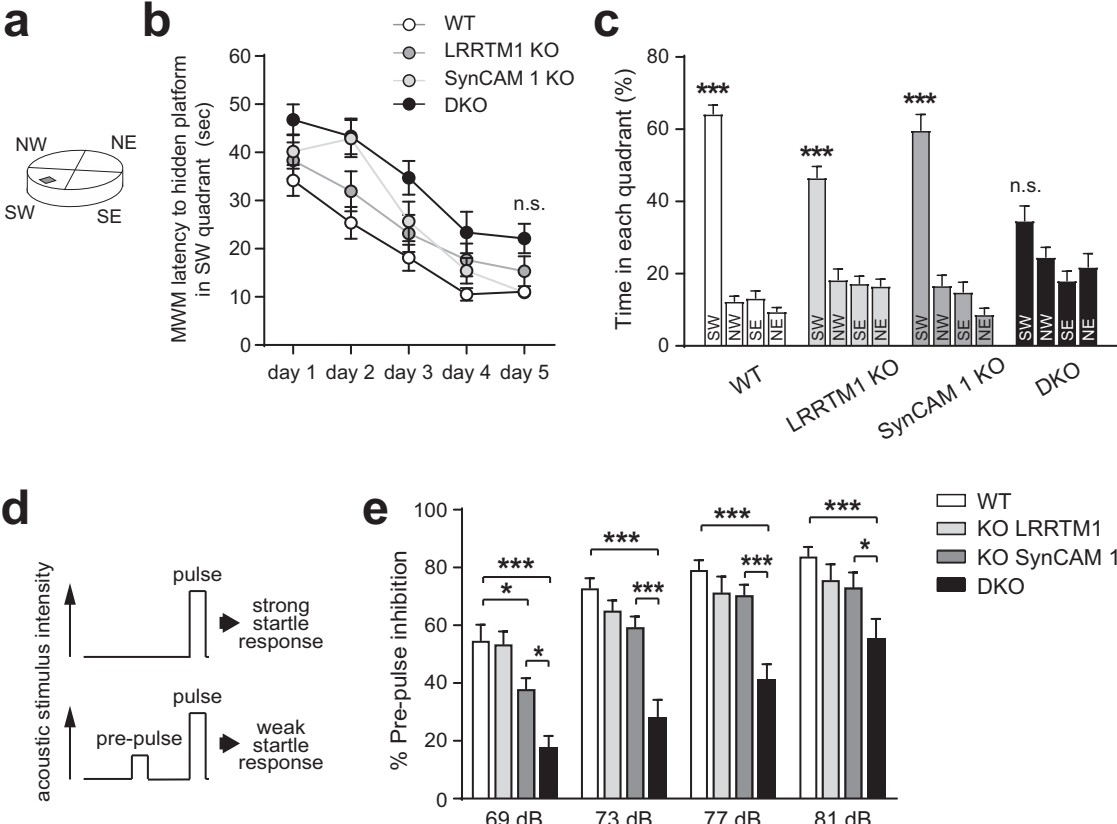

**Fig. 7 | Combined loss of LRRTM1 and SynCAM 1 impairs cognition-relevant behaviors. a** Spatial memory was tested in the Morris water maze divided into four quadrants, with the SW quadrant containing the escape platform. **b** Latency time to the target the SW quadrant in the Morris water maze, a measure of spatial memory acquisition, is indistinguishable between all genotypes after 5 daily training sessions. (Two-way ANOVA with Tukey's post hoc test; $n = 14$ WT, 14 LRRTM1 KO, 10 SynCAM 1 KO, 12 DKO mice). **c** Impaired memory processing in DKO mice. After Morris water maze training, time spent in the target SW quadrant vs. other quadrants was measured in a probe trial. Unlike WT and single KO mice, DKO mice exhibit no learned preference. (Two-way ANOVA with Bonferroni test for multiple comparison of means; $n = 14$ WT, 14 LRRTM1 KO, 10 SynCAM 1 KO, 12 DKO mice). **d** Design of prepulse inhibition (PPI) experiments. PPI is based on the startle response elicited by a loud sound (top). If preceded by a quiet prepulse sound, startle response is reduced in WT mice (bottom). **e** DKO mice fail to exhibit robust PPI while LRRTM1 KO and SynCAM 1 KO mice displayed normal PPI at all three louder prepulse sounds like WT mice. (Two-way ANOVA with Tukey's post hoc test; $n = 10$ WT, 9 LRRTM1 KO, 10 SynCAM 1 KO, and 9 DKO mice). *$p < 0.05$, ***$p < 0.001$, n.s., not significant. Error bars show Standard Error of the Mean. Source data for panels **c** and **e** are provided as a Source Data file.

0.3 µM Aprotinin) in a 2.0 ml glass-Teflon homogenizer with 10–15 strokes using a Glas-Col Tissue Homogenizer (Cole Palmer, Vernon Hills, IL) at 800 rpm. After keeping an aliquot, the homogenate was centrifuged at 800 g for 10 min at 4 °C to remove nuclei and large debris as pellet P1. Supernatant S1 was collected and centrifuged at 9,200 g for 15 min at 4 °C to obtain a crude synaptosomal fraction as pellet P2. This was resuspended in homogenization buffer (80% of initial volume used) and centrifuged at 10,200 g for 15 min at 4 °C to obtain a washed synaptosome fraction P2′. This was resuspended in suspension buffer lacking sucrose and stored at −80 °C.

Co-immunoprecipitations were performed from HEK293T cells individually transfected with pCAG HA-rat Nrxn1α (Addgene #58266), pCMV5 NL14 encoding rat Neuroligi-1[81], or pCAGGS mSynCAM 1 encoding mouse SynCAM 1[82]. 36–48 h post-transfection, cells were harvested in cold PBS and pelleted at 1000 g for 5 min at 4 °C. Pellets were washed in PBS and resuspended in lysis buffer (50 mM Tris pH 7.4, 150 mM NaCl, 5 mM EDTA, 1% Triton X-100, 1 mM PMSF, 1.5 µM Pepstatin A, 2.1 µM Leupeptin, 0.3 µM Aprotinin). Lysates were rotated for 1 h at 4 °C and cleared by centrifugation at 14,000 g for 20 min at 4 °C. Supernatants were collected, and protein concentrations were measured. 50 µg lysate from each transfection condition was used to prepare input samples containing HA-Nrxn1 alone, Nlgn1 alone, HA-Nrxn1 + Nlgn1, SynCAM 1 alone, or HA-Nrxn1 + SynCAM 1. Input samples were pre-cleared with 20 µl Protein G-Sepharose beads

(Invitrogen) for 1 h at 4 °C. Pre-cleared samples were incubated with 2 µl rabbit anti-HA antibody (Cell Signaling Technology Cat# 3724, RRID:AB_1549585) for 2 h at 4 °C followed by 1 h incubation with 30 µl Protein G-Sepharose beads. Beads were washed four times with ice-cold lysis buffer at 0.1% Triton X-100. Immunoprecipitated proteins were eluted with SDS-PAGE sample buffer.

Protein concentrations were measured using the Pierce BCA Protein Assay (Thermo Fisher Scientific). For SDS-PAGE, samples were reduced with sample buffer containing 5% β-mercaptoethanol at 95 °C for 5 min. A total of 15 µg protein was loaded per lane unless indicated otherwise. For immunoblotting, membranes were blocked with TBS-Tween (TBST) containing 10% milk powder/5% Normal Horse Serum (NHS), except for blots to be probed for LRRTM1, which were blocked with TBST/3% BSA and blots to be probed for Neurexins, which were blocked using TBST/1% milk to reduce background. Primary antibody incubations were overnight at 4 °C in TBST/5% milk/2.5% NHS, TBST/ 1.5% BSA (LRRTM1) or TBST/1% milk (Neurexin). For immunoblot detection of Neurexins, gels were transferred on PVDF membranes. β-Neurexin bands were quantified between 60 and 100 kDa based on the loss of immunoblot signal in this molecular weight range in triple Neurexin KO mice[83], the detection of β-Neurexins in knock-in mice[49], and the ligand-based enrichment of β-Neurexin isoforms[32]. AlexaFluor-conjugated conjugated secondary antibodies were applied at 1:1000 for 60 min at RT. Scans were performed on an Odyssey Imaging

System (Li-Cor, Lincoln, NE) and a FluorChem M System (ProteinSimple, San Jose, CA) and acquired using AlphaView software.

## Mouse lines

Constitutive SynCAM 1 KO mice had been shared by Dr. T. Momoi (National Institute for Neuroscience, Tokyo)[84]. Constitutive LRRTM1 KO mice were previously described[25]. Neurexin 1/2/3 [fl/fl] mice were described previously[85]. LRRTM1 and SynCAM 1 KO mouse lines were backcrossed for more than 10 generations to C57BL/6NCrl mice (Charles River) prior to beginning of the study and were maintained on this background. LRRTM1 and SynCAM 1 KO mice were bred as heterozygotes to generate WT and KO littermates for comparison. To obtain DKO mice and single LRRTM1 KO littermates, mice heterozygotic for the SynCAM 1 KO allele and homozygotic for the LRRTM1 allele were bred.

## Neuronal cultures

Dissociated primary neurons were prepared in HBSS as described[52] from hippocampi of Neurexin 1/2/3 [fl/fl] mice[85] of either sex derived from timed-pregnant dams at E17.5. After 0.25% trypsin and trituration, cell suspensions were plated onto 18 mm glass coverslips (Menzel-Glaeser) which were coated with poly-L-lysine (Sigma-Aldrich) at a density of 55,000 cells/coverslip. Coverslips were inverted onto a 70–80% confluent monolayer of astrocytes grown in 12-well plates (Falcon), after 4 h at 37 °C in plating medium (MEM, 10% horse serum, 0.6% glucose, 1 mM sodium pyruvate). They were incubated in neurobasal medium supplemented with B27, 0.5 mM glutamine, and 12.5 µM glutamate. After 3 div, media were refreshed with neurobasal medium supplemented with B27, 0.5 mM glutamine, and 5 µM cytosine arabinoside (AraC). Cultures were maintained at 37 °C in a humidified incubator with an atmosphere of 95% air and 5% $CO_2$.

## Lentivirus production and transduction of primary hippocampal neurons

For deletion of Neurexins in cultured neurons, we used lentivirus expressing either active or inactive Cre recombinase (Cre/ΔCre)[86]. Recombinant lentiviral particles were produced in HEK293 cells (passage 2–10; $5 \times 10^6$/10 cm dish). Cells were transfected using Lipofectamine 2000 (ThermoFisher) with pRSV-REV, pMDLg/gRRE, and pVSVG plus as separate lentiviral plasmid FSW-NLS-GFP-Cre or FSW-NLS-GFP-ΔCre[86]. 30 min before transfection, cells were gently washed with PBS (RT) and 8 ml of Opti-MEM + 10% Fetal Calf Serum (FCS) were added per 10 cm-dish. Per dish 200 µl transfection mix in total were prepared by mixing first 60 µl Lipofectamine 2000 and 40 µl plain Opti-MEM in one tube and 4 µg each of pRSV-REV, pMDLg/gRRE, pVSVG, and 12 µg of Cre or ΔCre plus plain Opti-MEM in another tube. After 5 min both volumes were combined and incubated in the dark for 30 min at RT. Then the transfection mix was added dropwise to the HEK293 tsA cells. Afterwards dishes were mildly shaken and put back to the incubator. 6 h following transfection, medium was replaced with DMEM containing 10% FCS. After 10–12 h, medium was exchanged by 10 ml of neuronal B27-medium and cells kept in the incubator for another 54 h. Medium was collected and centrifuged (500 g, 10 min, 4 °C). The supernatant was aliquoted, snap-frozen at −80 °C, and used within 4 weeks of freezing.

At 4 div, neuronal cultures were infected with lentivirus by adding 155 µl of viral supernatant per well of a 12-well plate. Lentiviruses either expressed GFP-Cre with enhanced nuclear localization driven by a synapsin promoter to generate conditional TKO neurons, lacking all neurexins, or an inactive variant of GFP-ΔCre to generate control neurons. Titer of virus particles was adjusted so that over 90% of neurons showed GFP expression in the nucleus.

## Immunocytochemistry

For labeling of hippocampal cultures, neurons were fixed at 21 div with 4% paraformaldehyde/4% sucrose for 10 min, washed with PBS, permeabilized with 0.3% Triton/PBS and blocked in 5% normal goat serum (NGS), PBS for 30 min, and incubated with primary antibodies overnight at 4 °C. Antibodies against Synapsin 1 and Neurexins were diluted in blocking solution for staining overnight at 4°. After washing, cells were incubated with secondary antibodies diluted 1:500 in 5% NGS/PBS for 1 h at RT. After PBS washes, coverslips were embedded in mounting medium (Dako).

## Immunohistochemistry

Immunohistochemical data were obtained from both male and female mice at P50-65. Mice were anesthetized with ketamine (100 mg/kg) and xylazine (10 mg/kg) in saline. Animals were transcardially perfused first with ice cold PBS and then with 4% PFA (in PBS, pH 7.4). Mice were perfused at P50-65. Brains were isolated and postfixed overnight in 4% PFA, then washed three times with PBS and stored in PBS (all at 4 °C). Brains were coronally sectioned at 50–70 µm using a vibrating microtome (Vibratome 1500, Harvard Apparatus, Holliston, MA). Sections were stored in PBS at 4 °C followed by immunostaining. Hippocampal and prefrontal cortex sections were washed in PBS and antigen retrieval was performed by submerging sections in sodium citrate buffer (pH 6.0) at 73 °C for 35 minutes. After allowing the sections to cool to RT, they were washed in PBS and non-specific antibody binding sites were blocked with 3% NHS and 0.1% Triton-X 100 (Sigma) in PBS for 2 h at RT. Primary antibodies used in double- and triple-labeling experiments were applied sequentially and diluted in 3% normal horse serum and 0.1% Triton-X 100 in PBS for incubation with the sections for 24–48 h at 4 °C. Secondary antibodies were applied in the same buffer overnight at 4 °C or for 2–3 h at RT. For SynCAM 1 immunohistochemistry, this protocol was modified, and brains were postfixed immediately after dissection in 4% PFA at 4 °C overnight and coronal sections of 60 µm were cut. SynCAM 1 antibodies were applied for 3 h at 4 °C to sections in a solution of 3% NHS and 0.05% Triton-X 100 in PBS, followed by other primary antibodies as above. After incubation with secondary antibodies, sections were washed in PBS and floated on slides in distilled water before coverslipping with mounting medium (CFM-3, Citifluor, Hatfield, PA, USA).

## Biolistic labeling

Biolistic labeling was performed in male mice at P56-63 as described[87,88]. Briefly, brains were removed from transcardially perfused mice, postfixed with 4% PFA for 1 h, and 300 µm sections were obtained on a Vibratome 1000 (Warner Instruments). Due to the thickness required of these slices, sections were cut starting from the most rostral to the middle parts of the mPFC up to its most caudal level as marked in Fig. 4 of ref. [37]. Tungsten particles were coated with 1,1′-dioctadecyl-3,3,3′,3′-tetramethylindocarbocyanine perchlorate (DiI; Invitrogen) and delivered using a Helios Gene Gun (BioRad, Hercules, CA). After incubation for 20 h at 4 °C in PBS, sections were fixed for 4–6 h and mounted.

## Confocal microscopy

Confocal images of immunostained tissue sections were obtained with a laser scanning confocal microscope (SP8; Leica Microsystems or Zeiss LSM 800) using a 63× oil immersion objective (1.3 NA) and Leica LAS software, or 40× water immersion objective (1.2 NA) and Zen software. Single optical sections of 0.8 µm thickness were imaged at a 2048 × 2048 resolution. For PV intensity quantification, Z-stacks of the entire section thickness were taken at 2048 × 2048 and step size of 5 µm. Identical image acquisition settings were applied for each group within an experiment. Images for immunohistochemical analysis of mPFC were acquired in cell layer II/III of primarily prelimbic area PrL

and cingulate area CG1 and also included the edges of PrL-IL and CG1-M2 as marked in Fig. 4 of ref. [37]. Images from hippocampus were acquired in the stratum radiatum of the CA1 region. Images after biolistic labeling were acquired from the prelimbic area of the most rostral part of the mPFC, cingulate Cg1, prelimbic and infralimbic areas of the middle parts of the mPFC, and cingulate area Cg1 of the most caudal level as in Fig. 4 of ref. [37]. During image acquisition, the researchers were blind to the genotype of each animal.

Imaging of immunostained neuronal cultures was performed with a confocal spinning disc Axio Observer-Z1 (Visitron) with an EMCCD camera (ImagEM 512 CCD, Hamamatsu), using 63× Plan-Neofluar oil-immersion objective (Zeiss).

## Analysis of imaging data

For quantification of vGlut1 and Homer synaptic puncta in immunostained sections, the machine-learning Intellicount algorithm[89] that identifies them in a threshold-independent manner was applied as described[90]. vGlut1 and Homer puncta channels were analyzed independently. Intellicount settings used were background removal factor 0.1, default threshold, "single channel/greyscale" and "synapse" options were selected, and a maximum size of puncta set to $2.5\,\mu m^2$. Resulting pre- and postsynaptic puncta traces produced by Intellicount were then stacked in ImageJ and scored for co-localization for synaptic alignment using the cell counter application.

SynCAM 1 fluorescence intensity in immunostained sections was quantified with ImageJ from optical sections of $1.0\,\mu m$ thickness acquired on a Zeiss LSM 800 confocal microscope. Size-corrected background was calculated, and the corrected total region fluorescence (CTRF) was determined. For PV intensity quantification, background subtracted maximum intensity projections were used. Mean grey value was measured in identified PV cell bodies that were defined as circular ROIs.

Images for quantification of DiI-labeled dendritic spines were obtained by confocal imaging setting stacks from top to bottom of dendrites with a z-step size set to $0.30\,\mu m$. Maximum projection images were obtained using ImageJ. Images were thresholded and binarized, and the ImageJ cell counter application was used for morphological analysis. Analysis of images after biolistic labeling was performed as described[88] and was restricted to secondary dendritic branches. We classified dendritic protrusions as thin spines, mushroom-type spines, stubby spines, and filopodia-like protrusions as described[45]. Briefly, spines with a head-to-neck diameter ratio of less than 4 were categorized as thin type, and if the head-to-neck diameter ratio was larger than 4, as mushroom, with classifications independent of neck length. The stubby category corresponded to spines that protruded only narrowly and lacked a detectable neck. The filopodia category was comprised of long protrusions without a clear diameter difference of neck and tip.

## In vivo electrophysiology

Electrophysiological data were obtained from both male and female mice. Recordings were performed on awake female and male mice, aged P35-P80, using spherical treadmill as described[91]. In total 4−7 days before the recording session, custom made titanium head-plate implants were cemented to the mouse skull. Animals were anesthetized with isoflurane in oxygen (2% induction, 1.0−1.8% maintenance), warmed with a heating pad at 38 °C and given subcutaneous injections of Buprenorphine SR (1 mg/kg) and 0.25% Bupivacaine (locally). Eyes were covered with Puralube (Decra, Northwich, UK). Scalp and fascia from Bregma to behind lambda were removed, and the skull was cleaned, dried and covered with a thin layer of cyanoacrylate (VetBond; 3 M, Maplewood, MN) before attaching the head plate with dental cement (RelyX, 3 M). The well of the head plate was filled with silicone elastomer (Reynold Advanced Materials, Brighton, MA) to protect the skull before recordings. Animals were single housed

after the implantation and monitored daily for signs of shock or infection. In total 6 days before the recording, the animals underwent daily 5−10 min handling sessions and 10−15 min sessions in which the animals were habituated to the spherical treadmill[92]. On the day of recording, the animals were anesthetized as above and small craniotomies (∼0.5 mm in diameter) with 18 G needles were made above left mPFC (1.5−2.3 mm anterior to Bregma, 0.2−0.3 mm lateral to midline) and cerebellum (for reference electrode). The brain surface was covered in 2−3% low melting point agarose (Promega, Madison, WI) in sterile saline and then capped with silicone elastomer. Animals were allowed to recover for 2−4 h. For the recording sessions, mice were placed in the head-plate holder above the free-floating ball and allowed to habituate for 5−10 min. The agarose and silicone plug were removed, the well was covered with warm sterile saline and the reference insulated silver wire electrode (A-M Systems, Carlsborg, WA) was placed in cerebellum. A multisite electrode spanning all cortical layers (A1x16-5mm-50-177-A16; Neuronexus Technologies, Ann Arbor, MI) coated with DiB (Biotium, Freemont, CA) was inserted in the brain through the craniotomy and slowly lowered until to the depth of 2−2.2 mm. the electrode was allowed to settle for 20−30 min. The well with the electrode was then filled with 3% agarose to stabilize the electrode and the whole region was kept moist with surgical gelfoam soaked in sterile saline (Pfizer, MA). 3 penetrations were made per animal to ensure proper sampling of the craniotomy. Recording sessions typically lasted 0.5−1 h. After the recording, mice were euthanized with an overdose of ketamine and xylazine and perfused with warm PBS and 4% PFA as described above. Brains were postfixed in 4% PFA for 1 h at RT followed by overnight postfixation at 4 °C, and subsequently sectioned at 40 μm using a vibrating microtome to validate the recording site based on DiB label from the electrode insertion tract and blood vessels and dura puncture as visual markers for the region.

## Electrophysiological data collection and analysis

Local field potentials (LFPs) were preamplified 10× (MPA8I preamplifiers; Multi Channel Systems MCS GmbH, Reutlingen, Germany) and then fed into a 16-channel amplifier (Model 3500; A-M Systems), amplified 200× and band-pass filtered 0.3−5000 Hz. The signals were sampled at 25 kHz using Spike2 and data acquisition unit (Power 1401-3, CED). Stationary and movement stages of animal behavior were separated using an optical mouse that tracked the movement of the Styrofoam ball (Spike2, CED). Only stationary, non-running stages were analyzed offline using Spike2 software (CED). For multi-unit analysis, spikes were extracted from band-pass filtered data using thresholds (mean + 3× standard deviation), and firing rate was extracted from Spike2 using 1 s bins. Data collection and analyses were performed blind to genotype or experimental group.

Single units were identified from band-pass filtered data using thresholds (mean + 3× SD) and template matching in Spike2. Isolated units were sorted using principal component analysis and plotted using the latency between the first peak and the trough to delineate the narrow and broad spiking units.

## MRI scanning

Mice were anesthetized with 25%/75% urethane/distilled water. For most mice, an initial bolus of 1.25 g/kg was given, divided equally into 3 doses given 5 min apart. Following the initial bolus an intraperitoneal line was surgically inserted and this was used to give additional urethane, if needed, until no toe pinch reflex was observed. The exact amount needed to suppress the toe pinch reflex varied based on the mouse 1.54 ± 0.22 g/kg (mean ± one SD), in line with previous work[93]. No significant genetic difference in dose was observed (ANOVAN, $p = 0.90$ SynCAM 1 gene, $p = 1.00$ LRRTM1 gene, $p = 0.86$ interaction). Anesthetized mice were placed in a custom 3D-printed holder inside a custom-built MRI surface coil. Mice had eye ointment applied to reduce stress, were monitored by rectal temperature probe, and were

heated by a water bath with temperature adjusted to maintain body temperature 36–38 °C. Following positioning inside the scanner, static magnetic field inhomogeneities were corrected using B0DETOX[94] (http://innovation.columbia.edu/technologies/cu17326_b0detox). Mice were imaged with a 9.4 T Bruker horizontal-bore spectrometer using a home-built $^1$H surface coil radiofrequency probe (1.2 cm diameter). Details of fMRI measurements for BOLD were discussed previously[95]. Briefly, BOLD signal was acquired with echoplanar imaging (EPI) with sequential sampling[96] using gradient-echo contrast. Resting state fMRI scans were collected for 2048 s at 0.5 Hz (25.6 × 12.8 mm FOV, 64 × 32 matrix, 8 slices, 1 s TR + reference scan every image, 13 ms TE, 1024 repetitions). Following functional imaging, an FSEMS anatomical image was acquired coplanar to functional data (20 × 10 mm FOV, 128 × 64 matrix, 24 slices, 2000 ms TR, 6 ms TE, 16 directions). Anatomical images were used in native space for comparison to functional images. One SynCAM 1 KO and two DKO mice died in the setup and thus had no functional images acquired but did have anatomical images acquired.

## MRI data processing

Functional data analysis was done for individual mice without a template brain, using MATLAB with BioImage Suite and SPM8 (Wellcome Trust Centre for Neuroimaging, http://www.fil.ion.ucl.ac.uk/spm/software/spm8/). GS was analyzed as described[97]. Briefly, each mouse's resting state run was divided into four segments of 512 s each. Data were filtered to 0.01 to 0.03 Hz[93] and analyzed without regression of the mean signal from the whole-brain mask, or 'global signal regression'[98,99]. The mean signal from the whole brain, or 'global signal' (GS) was saved for each segment. The standard deviation was calculated on each segment and the mean taken over all kept segments for each mouse. Statistics were performed on these standard deviations to determine differences in GS between groups. As motion has been implicated in the BOLD-fMRI GS[100], motion parameters were further tested. First, standard deviation and range (maximum minus minimum position shift) of motion time series were calculated for the same time periods on which functional connectivity was collected and averaged per-mouse to give the same data points as the GS data. Three-factor ANOVAN was run on motion parameters with SynCAM 1 presence, LRRTM1 presence, and specific direction as the factors. This was run separately for rotation and translation, and separately for standard deviation and range. No results were found with $p < 0.05$ for any genetic difference, though there was a slight direction preference (unrelated to genetics) for translation and standard deviation ($p = 0.03$–0.09 direction, $p = 0.46$–0.67 SynCAM 1 gene, $p = 0.31$–0.79 LRRTM1 gene, $p = 0.09$–0.14 direction/SynCAM 1 interaction, $p = 0.49$–0.74 direction/LRRTM1 interaction, $p = 0.13$–0.25 SynCAM 1/LRRTM1 interaction). Second, resting state data analysis was re-run with regressing the six motion parameters from raw data following image registration and before filtering. This produced nearly identical results as without regression of motion parameters (N-way ANOVA, $p = 0.0003$ SynCAM 1 gene, $p = 0.11$ LRRTM1 gene, $p = 0.30$ interaction; one-tailed, equal variance, two-sample t-test for DKO > SynCAM 1 KO, $p = 0.018$). This suggests motion was not a covariate within GS. Differences in GS were therefore not due to genetic differences in motion.

For each metric, if multiple runs or measurements existed for each mouse, these were averaged. To control for group effects and against multiple comparisons, N-way analysis of variance (ANOVAN) was used with each independent measurement being one mouse, and independent variables being presence of SynCAM 1 knockout (positive for SynCAM 1 KO and DKO groups) and presence of LRRTM1 knockout (positive for LRRTM1 KO and DKO groups). This provided p values for significance of the SynCAM 1 gene, the LRRTM1 gene, and interaction. If either KO was significant, a post-hoc t-test was performed between members of that group, i.e. LRRTM1 KO and DKO for LRRTM1 and SynCAM 1 KO and DKO for SynCAM 1.

## Behavioral studies

Behavioral tests were performed using cohorts of male mice at P56-63. Cohorts were tested in series from least stressful to most invasive experiments. Results from open field, social interactions and PPI were obtained from cohorts subjected to these tests in this order. Results from Rotarod and elevated plus maze were obtained from separate cohorts subjected to these tests in this order. Three cohorts were tested in the Morris water maze and not used for other experiments. Open field, social interactions, and PPI experiments were performed with mice housed on a reverse light 12 h dark/light cycle so that tests were conducted in the wake phase. Tests of motor coordination, elevated plus maze, and Morris water maze were performed with mice housed on a regular 12 h light/dark cycle.

**Open field.** Locomotor activity was measured in an open field using using a SmartFrame System and MotorMonitor software (Kindler Scientific, Poway, CA). Mice ($n = 11$) were placed in the middle of a 50 cm × 50 cm × 20 cm Plexiglas enclosure with opaque walls and allowed to explore it freely for 60 min. Walk speed was measured by the tracking software and distance traveled during a 60 min period was binned in 5 min intervals.

**Social Interactions.** The test was performed using the EthoVision v10.0 video tracking system (Noldus, Leesburg, VA) in 12–16 mice per genotype, using a three-box design[101,102]. The time mice spent in any of the three compartments during two experimental sessions of 10 min each, including indirect contact with the unfamiliar mouse, was recorded. The unfamiliar mouse corresponded to a juvenile mouse (6–7 weeks old) with the same genetic background and gender and without prior contact with the subject mouse.

**Motor coordination.** Motor coordination was evaluated using a Rotarod apparatus (Columbus Instruments, Columbus, OH) set at a baseline speed of 2 rpm, with acceleration at 0.2 rpm/s. Latency for the animal to fall off the rotating drum was recorded. Each session included five consecutive trials, with the average fall latency calculated from the best three trials per animal.

**Elevated plus maze.** Elevated plus maze studies were performed as described[103]. The apparatus consisted of two open and two closed arms of each 30 cm × 5 cm with edge heights of 4 mm and 15 cm, respectively. Arms were connected by a central platform of 5 cm × 5 cm. Floors were white Plexiglas. The apparatus was elevated 50 cm. Each mouse was placed in the central platform facing an enclosed arm and allowed to explore the maze for 5 min. The four-paw criterion was used to qualify for arm entry. Time spent and distance traveled in the open versus closed arms was recorded manually.

**Prepulse inhibition.** Prepulse inhibition (PPI) was performed using the Startle Reflex System and Startle Monitor software (Kindler Scientific) as described[104]. Initially the mice were subject to 6 pulses (120 dB) trials for habituation after 5 min background white noise (70 dB) as acclimation period. The delay between the prepulse and the pulse during trials was fixed at 80 msec. The percent inhibition was calculated as PPI = 100× [(pulse alone–prepulse)/pulse alone].

**Morris water maze.** Morris water maze studies were performed as described[35,105]. Mice received four training trials per day during their light cycle for 5 days ($n = 10$–14). The intertrial interval was 5 min. Animals were placed in a water-filled circular white plastic tank (diameter 100 cm, water temperature 21–22 °C). A clear plastic platform (10 cm × 10 cm) was submerged 0.5 cm and placed in the same location in the tank over the training days. For analysis, the tank was divided into four quadrants, with animals starting one trial in each quadrant on all training days. The order of starting quadrants was randomized, and

mice were placed facing the tank's edge. Salient visual cues of different shape and color were mounted on the tank wall. Path length, time spent in each quadrant, and latency to find the platform were measured by a SMART video tracking system (Panlab, Harvard Apparatus, Holliston, MA). Mice that did not find the platform within 60 s were manually placed onto the platform. All animals were allowed to remain on the platform for 15 s. On day 6, the probe trial was performed. The platform was removed, the mice were placed in the middle of the tank and allowed to swim for 60 s. Time spent in each quadrant was recorded. On day 7 the platform was moved to a different quadrant, marked with a flag and a block of three swims to the visible platform was conducted.

## Statistical analysis

All imaging data acquisitions for quantitated analyses and their quantitation were performed with the researcher blind to the conditions. Analysis was performed using GraphPad Prism 9.4.1 (Graph Pad Software, La Jolla, USA). Statistical analyses were performed as indicated in the figure legends, with errors corresponding to the standard error of the mean. Boxes in whisker plots extend from the 25th to 75th percentiles and the middle line is plotted at the median, with whiskers to maxima and minima of all data. * denotes $p < 0.05$; **$p < 0.01$; ***$p < 0.001$.

Regarding dendritic protrusion densities, we tested for genotype-dependent differences by comparing each protrusion type (thin, mushroom, and stubby spines, and filopodia-like) separately using one-way (genotype) ANOVA. Two-way ANOVA was not performed because these four types of dendritic protrusions can dynamically transition into each other and were not considered to be independent. Data were analyzed by fitting a mixed model and Tukey's test was used to detect differences.

For MUA firing rate data, the Kruskal-Wallis test was performed because the data was not normally distributed in all groups. Specifically, there did appear to be two populations in the KO lines, while WT data was normally distributed. This non-normal distribution of firing rate data could reflect that the loss of LRRTM1 and SynCAM 1 has a more pronounced phenotype in distinct cell types. This would be consistent with their cell-type specific roles in synapse specification.

## Reporting summary

Further information on research design is available in the Nature Portfolio Reporting Summary linked to this article.

# Data availability

A source data file for the figures is provided with this paper, including unprocessed scans of immunoblots and graphed data points shown in the main paper and the supplemental information. Because of their size, raw microscopy images, electrophysiological recordings, and MRI scans obtained and analyzed in this study will be made available upon request to the corresponding author.

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

## Acknowledgements

We thank Dr. Amy Arnsten, Dr. Alex Kwan, and the members of the Biederer lab for discussions, Dr. Daniele Repetto and Vivien Rieping for help with neuronal cultures and Neurexin antibody controls, Isabelle Witteveen for assistance with confocal imaging and analysis, Dr. Andrew Tarr and the Behavioral Core of the Tufts Center for Neuroscience Research for support, Dr. Marina Picciotto for access to behavioral equipment, Dr. Takashi Momoi for the SynCAM 1/RA175 KO mouse line, and Dr. Feng Dai for expert advice with biostatistics. Dr. Ananda Ghosh established the LRRTM1 KO line and Yuling Lei provided technical assistance in early project stages. This work was supported by the Deutsche Forschungsgemeinschaft (SFB1348, TP A03, to M.M.), NIH grant R35 NS097283 (to S.M.S.), NIH grant P30 NS052519 (to F.H.), and NIH grants R01 DA018928 (to T.B.) and R01 MH119826 (to T.B.).

## Author contributions

K.P.d.A., A.R., D.C., K.W., and T.B. conceived approaches; K.P.d.A. performed and analyzed biochemical and histochemical studies and behavioral experiments; A.R., electrophysiological and immunohistochemical

studies; D.C., biochemical studies; K.W., immunohistochemical studies; G.J.T., B.G.S., E.T.C.L., fMRI data acquisition and analysis; A.Ro. and M.M., conceived, performed and analyzed immunocytochemical studies; S.M.S. provided LRRTM1 KO mice, F.H. supervised fMRI studies, and T.B. supervised K.P.d.A., A.R., D.C., and K.W., and wrote the manuscript.

## Competing interests

The authors declare no competing interests.
