## [Peer Review File · Nature Communications]

Concerted roles of LRRTM1 and SynCAM 1 in organizing prefrontal cortex synapses and cognitive functionsREVIEWER COMMENTS

Reviewer #1 (Remarks to the Author):

This manuscript from the Biederer lab investigates the potential co-operativity between 2 synaptic adhesion molecules, LRRTM1 and SynCAM1 in the rodent prefrontal cortex (PFC). This is interesting, due to the numerous different adhesion molecules that coordinate synapse organization and number in the brain, but less is known about how they might work together, and how much redundancy there is in their function. Importantly, for this work they focus on the PFC, the site of disruption for schizophrenia, and how LRRTM1 and SynCAM1 might work together for proper synapse formation in this area, neural function and behavior. The authors show that loss of LRRTM1 in the brain results in compensatory increases in expression of SynCAM1 and alpha-neurexins in PFC. The LRRTM1 KO animals have few deficits, but a double KO of both LRRTM1/SynCAM1 exhibit deficits in several cognitive and schizophrenia related behavioral domains. In addition, these mice display alterations in brain activity and neuron firing, together suggesting that both of these proteins need to be lost for proper brain function.

Overall, this paper combines an impressive range of in vivo and ex vivo techniques to try to address an important question concerning trans-synaptic organizing proteins and how they might work together in controlling synapse number. There are some interesting observations, the data seem solid and it is fairly well-written. However, there are some issues that need to be resolved before publication including some critical holes in the analysis, a need for further in depth mechanistic insight and a less disjointed manuscript organization.

Main points

1. The organization of the story is disjointed (going back and forth between the molecular composition of synapses and in vivo readouts in whole animals, then back to the 'molecular remodeling') making it hard to follow the message. It may be better to present the molecular changes/ synapse aberrations together, followed by the behavioral in vivo output of the synapse disruptions.

2. As the hypothesis is that this compensation is happening specifically in PFC, there are some key outstanding questions that should be answered to confirm this and support the existing data.

-An obvious question seems to be if there are significant differences in LRRTM1, SynCAM1, NLs and Nx in PFC synapses compared to other brain regions (i.e. are LRRTM1 and SynCAM1 particularly enriched there compared to hippocampus, whole forebrain etc.) If LRRTM1 was more highly expressed in PFC this would contribute to the argument that it might contribute to schizophrenia-like phenotypes.

-The mix of brain regions used for the biochemistry is inconsistent, making it difficult to make strong conclusions e.g. Fig 1. C, D use SPM from frontal cortex, but Supp. Fig 1A,B use SPM from forebrain. Due to the cooperativity of NL1 with LRRTM1, it is important to know if NL1 (and neurexins) are upregulated in SPM of frontal cortex.

3. The ex vivo imaging and analysis are nicely done (especially the use of machine learning to define a synapse), although they require further controls and there are some key experiments missing that would significantly strengthen the conclusions of the paper:

- Figure 1A: Synapse density is measured using the association of vGlut1 and Homer1 to define a synapse, revealing a strong reduction in excitatory synapses. The analysis is incomplete and needs to be expanded: are there changes in synapse area/volume? VGlut1 and homer should be analyzed independently for density/area, and the % of Vglut1 with/without homer (and the reverse) should be quantified (as reported in Fig 5 e,f- from the images it looks as if many Vglut1 positive synapses that do not have post-synaptic densities, and this is not shown in the data).

-Considering Fig 1A forms part of the main crux of the story, analysis of other excitatory markers, such as PSD-95/AMPA receptors, and for inhibitory synapses (particularly as E/I coordination and interneuron loss in this region are strongly implicated in Schizophrenia) should also be performed to get a sense of the extent of synapse disruption.

-A direct comparison of synapse number between SynCAM1 and LRRTM1 single KO mice and the DKO mice is missing, although the imaging has been done (but analysis not included). Fig 5e,f reports Homer puncta colocalizing with vGLUT, but not synapse numbers. This would support the claim that KO of both proteins augments the effects on synapses.

- The imaging of SynCAM1 in Supplementary Fig 1 C is unconvincing, and not quantified. This should be repeated as in Fig 1A with quantification and analysis of SynCAM1 with excitatory synaptic markers.

4. One of the conclusions of the paper is that 'the adhesion molecules LRRTM1 and SynCAM 1 are interdependently expressed at synapses in the mPFC'. More imaging data are needed to fully support this statement. Although the biochemical data presented suggest this is the case, the presence of these proteins at synapses in the respective knockdown conditions should be analyzed. A key mechanistic question (acknowledged by the authors) is whether LRRTM1 and SynCAM1 are actually in the same synapses, and exactly how their actions are coordinated. Further, although NL1 upregulation in the LRRTM1 KO is mentioned, presumably NL1 is having a significant compensatory role in these mice- this needs to be addressed more fully (e.g. labeling with NL1 in the KO mice).

5. The reduction of firing rate in of PFC layer II/III neurons is compelling, but mechanistic insight into what is driving this change in firing is lacking (and would link the alterations in synapses to the in vivo observations). Is it primarily due to the loss of post-synaptic sites/spines? Or is the the observed increase in NX cluster size causing problems with synaptic release? Is inhibition altered? Ex vivo electrophysiological analysis of the DKO mice would answer these questions.

6. Throughout, it is hard to reconcile the link between the initial findings of LRRTM1 loss in PFC through to the behavior with schizophrenia-like phenotypes. It is not surprising that substantially disrupting glutamatergic circuits in in the brain will produce those behavioral phenotypes (even if they are associated with the PFC). Currently, the authors do not provide compelling evidence linking LRRTM1 loss, SynCAM1 compensation and schizophrenia-like behaviors. Given the loose genetic association of LRRTM1 with schizophrenia, and no evidence to suggest that synapse loss specifically in PFC leads to the behaviors observed in these constitutive KO mice, the link with schizophrenia becomes somewhat tenuous. Addressing whether the PFC of DKO mice exhibit other cellular and circuit characteristics (eg. loss of PV+ interneurons) of schizophrenia pathology might go some way to addressing this issue.

Reviewer #2 (Remarks to the Author):

In this paper, Dr. Biederer's group has shown the interdependent synaptic expression of LRRTM1 and SynCAM1 in the medial prefrontal cortex (mPFC) and their concerted functions for regulating schizophrenia-relevant behaviors, excitatory synapse number and so on. Specifically, their immunohistochemical and biochemical analyses show that the mPFC in LRRTM1 KO mice has a significant reduction of excitatory synapse density and a significant increase in synaptic expression of alpha-neurexin, NLG-1 and SynCAM1 as possible compensatory upregulation. Also, they found an increase in LRRTM1 synaptic expression in SynCAM1 KO synaptic membrane fraction, suggesting an interdependent expression of LRRTM1 and SynCAM in the mPFC. Such interdependent expression of them is not detected in the hippocampus. Given the interdependency between LRRTM1 and SynCAM1, the authors next characterized single global knockout (KO) mice of LRRTM1 or SynCAM1 and double KO (DKO) mice for them by behavioral tests, neural activity measurements, dendritic spine analysis and immunohistochemical analysis. They found that DKO mice, but not each single KO mouse line, show 1) memory impairment in the probe test of Morris-Water maze test, 2) an impaired attentive processing in the prepulse inhibition test, 3) social deficit in the three chamber test, 4) an alteration in brain and neuronal activity in the mPFC in fMRI and in vivo recording, 5) more additive reduction of dendritic spines in the layer II/III pyramidal neurons in the mPFC, but not in the hippocampal CA1 pyramidal neurons, 6) less co-localization of Homer puncta with VGLUT1 puncta

and 7) an alteration of neurexin puncta size. Thus, through anatomical, biochemical, functional and behavioral assessments, the authors show many consistent results to claim the concerted effects of LRRTM1 and SynCAM1 KOs on multiple levels from synapse level to behavioral one. This is a strong point in this paper, thereby it may have a certain degree of potential to attract many readers of the journal. On the other hand, I think that this paper has three major weak points, which limit appropriate data interpretation and dampen potential scientific impacts of this study, to be improved by additional experiments and revisions. My major and minor comments are described below.

Major comments

1) This paper has no data regarding the assessment of synaptic functions. Although they performed fMRI imaging and in vivo recording, these allow them assess brain activity and neuronal activity but not directly assess the function of synapses. Due to the lack of the data about synapse functions, some anatomical data remain confusing for me to consider the author's claim about the concerted role of LRRTM1 and SynCAM1 on some behaviors. Especially, Fig.1a shows drastic reduction of excitatory synapses in the mPFC in the single KO of LRRTM1, and Fig. 5f shows significant reduction of Homer colocalization with VGLUT1 in the mPFC compared to WT littermates. However, LRRTM1 single KO mice show comparable behaviors to WT in all performed behavioral tests. One possible reason is that the remaining excitatory synapses in LRRTM1 KO mice would be more functional than those in WT littermates to compensate the reduction of excitatory synapse number. However, there is no electrophysiological data to explain such gap between anatomical deficits of excitatory synapses and no behavioral phenotypes. Further, Both LRRTM1 single KO mice and DKO mice show reduction of dendritic spines (Fig. 5a,b) and impaired colocalization of Homer with VGLUT1 but show opposite phenotypes in firing rate of mPFC pyramidal cells (Fig. 5d). This apparent discrepancy would be due to different phenotypes in synaptic functions between these two mouse lines. (e.g. again, synapses in LRRTM1 KO are more functional than those in WT due to SynCAM1 upregulation, whereas synapses in DKO are less functional than in WT because of loss of the SynCAM1 upregulation as well as its default expression.). Thus, the assessment of synapse function by electrophysiological experiments is indispensable to achieve more convincing data interpretation from molecular, anatomical level to behavioral level.

2) Excitation/inhibition (E/I) imbalance in the mPFC is known to be one of key pathophysiological conditions underlying social behavioral deficits. However, in this paper, the authors have not analyzed anything regarding inhibitory synapse. Indeed, previous studies have suggested that LRRTM1 and SynCAM1 represent excitatory, but not inhibitory, synapse localization and trigger excitatory, but not inhibitory, synaptogenesis. However, the reduction of excitatory synapses primarily induced by LRRTM1 single KO and LRRTM1/SynCAM1 DKO might result in some changes in GABAergic inhibitory synapse number and/or function as a secondary effect presumably through activity-dependent homeostatic mechanism and/or compensatory mechanism that normalizes the E/I imbalance. Further, the author's data show LRRTM1 KO-induced upregulation of synaptic alpha-neurexins, which have synaptogenic activity to induce inhibitory, but not excitatory, postsynaptic differentiation (J Biol Chem (2008), 283:2323-34). The further analysis of inhibitory synapses to elucidate whether E/I imbalance in the mPFC occurs or not in each KO mouse line is essential to improve data interpretation of Fig.3 c.d (Social behavioral test) and Fig 4 (Brain activity and neural activity phenotypes).

3) This paper has no experimental evidence regarding mechanisms underlying interdependent synaptic expression of LRRTM1/SynCAM1 in the mPFC. Therefore, the current paper is relatively descriptive with just listing many phenotypes of single KO and DKO mice. Also, considering several previous studies has already shown interdependent expression and/or concerted effects of two different types of synaptic organizers with providing mechanistic insight (e.g. Slitrk3/NLG2 in Neuron (2017), 96:808-826, LRRTMs/NLG1/3 in PNAS (2011), 108: 16502-9 and J Cell Biol (2011), 194(2):323-34), I think that it would be necessary for this paper to add some experimental data that can provide mechanistic insight into how LRRTM1 and SynCAM1 interdependently regulate the expression of each other in the mPFC.

In the discussion section, the authors discuss a slot model as one of possible mechanisms. It has been reported that presynaptic scaffold CASK binds to SynCAM1 and neurexin, an LRRTM1-binding

partner. Thus, it would be interesting to test whether SynCAM1 and LRRTM1 compete with each other for CASK binding and whether the loss of one increases scaffold slots available for the other by losing the binding competition. Especially, given that the authors show altered expression and puncta size of neurexin in the mPFC of the DKO mice, it would be crucial for this paper to characterize CASK, a common binding partner for neurexin (LRRTM1-binding partner) and SynCAM1.

In addition, is it possible whether SynCAM1 can bind to neurexin itself? If yes, is it possible whether SynCAM1 and LRRTM1 compete with each other for neurexin-binding? The authors mention that SynCAM1 does not bind LRRTM1 (refs 36,37), but there is no information or discussion about SynCAM1/Neurexin binding.

Minor comments:

4) Although the authors wrote that “SynCAM1 is predominantly postsynaptic (line 114)”, the phrase of “the postsynaptic adhesion molecules LRRTM1 and SynCAM1” in line 394-395 should be an over-statement and biased because SynCAM1 is also expressed at presynaptic site to act as a presynaptic adhesion molecule with homophilic trans-interaction with postsynaptic SynCAM1. Also, the authors use global KO of SynCAM1, which cannot allow them to discriminate the role of postsynaptic SynCAM from that of presynaptic one. More careful data interpretation and unbiased discussion are necessary.

5) The authors use anti-neurexin antibody (Millipore, ABN161), which can recognize both alpha- and beta-neurexins. However, they quantified only alpha-neurexin. They need to quantify beta-neurexins too.

6) How about synaptic expression of alpha- and beta-neurexins and neuroligin-1 in DKO mice? These results would be important to discuss whether compensatory role of SynCAM1 under LRRTM1 deletion is dependent on or independent from these synaptogenic molecules and would be also helpful to consider possible mechanisms underlying interdependent synaptic expression of LRRTM1/SynCAM1 in the mPFC.

7) In Fig. 6b, the representative image of PFC SynCAM1 KO shows the weakest neurexin immunoreactivity, which is not consistent with the quantitative data. It should be replaced with a better representative one.

In addition, it is unclear what the enlarged neurexin puncta size in the DKO mice mean, enlarged presynaptic terminals, an increase in extra-synaptic neurexin puncta, or anything else? To answer this question, co-immunolabeling of presynaptic marker (e.g. Bassoon or Vglut1) with neurexin like Fig 6a and the quantification of puncta density and size of synaptic neurexin and non-synaptic neurexin would be necessary.

8) The authors used parametric tests (e.g. student's t-test for two groups and ANOVA for multiple groups) for all statistical except Fig 4d but used Kruskal-Wallis ANOVA (non-parametric test) for only Fig.4d. However, there is no description about the criteria in which parametric or non-parametric test was used (e.g. assumption of normality). In the statistics section in the Reporting Summary, they selected “N/A” in “A description of any assumptions or corrections, such as tests of normality and adjustment for multiple comparisons”, suggesting that they have not tried assumption of normality even when they used a parametric test. However, if the data are not normally distributed, using parametric test is not appropriate and thereby results in misleading conditions. Therefore, the authors need to revalidate if they actually use an appropriate statistical test for each data set by assuming data normality and homoscedasticity.

Reviewer #3 (Remarks to the Author):

The manuscript of deArce et al., addresses the important issue of how synaptogenesis is achieved through specific transsynaptic molecular complexes. The authors demonstrate using mouse genetic models and biochemical studies that two postsynaptic transmembrane proteins linked to schizophrenia LRRTM1 and SynCAM 1, work in concert to achieve excitatory synapse formation in the medial prefrontal cortex. A compensatory role of LRRTM1 and SynCAM I was demonstrated by detailed analyses of dendritic spines and excitatory synapses, neuronal activity, and behaviors, made possible by the generation of novel double knockout (DKO) mice. The experimental approach and techniques used in this paper were impeccably conducted and involved advanced techniques including fMRI to measure brain activity by resting state BOLD signals, electrophysiology in live mice, and spine analysis using a machine learning method for quantification. Experiments were rigorously designed in terms of numbers of measurements, statistical comparisons, and presentation of most data in scatter plots. The study showing interdependent regulation of expression of presynaptic Neurexin and postsynaptic LRRTM1 and SynCAM 1 is convincing but somewhat limited in lacking a mechanism (although the authors address this in the Discussion). Overall, this paper provides exciting new information illustrating that "many hands make light work" in excitatory synapse development. Specific points that should be addressed are as follows:

1. Data in Figure 1 showing LRRTM1 control of excitatory synapse density and expression should report the exact P values for significant differences among the genotypes, rather than $p < 0.05$.
2. Tests of social behavior as shown in Figure 3 demonstrate that DKO mice are impaired for sociability in the 3 chambered test for object versus stranger mouse. For a more rigorous second order test, the authors should consider adding the more complex 3 chambered test for social novelty preference involving a second unfamiliar mouse.
2. In the electrophysiology experiments of Figure 4 c,d, was a Kruskal-Wallis Anova performed because there were 2 populations of firing rates within the mutant data? If so, what would be an explanation for the different populations?
3. In the spine density results of Figure 5 explain why there is a separate category for large head mushroom spines, as generally these are included within the standard mushroom spine category. Since spines dynamically "morph" among the different morphologies, mushroom, stubby and thin spines are not truly independent variables. Thus, Anova may not be applicable; instead multinomial regression analysis would be more appropriate for comparing differences among genotypes. The authors should consult with a biostatistician about this. Minor point, density is misspelled on the y-axes of b and d.

Response to the Reviewer Comments on NCOMMS-20-39222, Concerted roles of LRRTM1 and SynCAM 1 in organizing prefrontal cortex synapses and cognitive functions

We would like to thank the Reviewers for their strong interest in this study and their insightful and constructive comments that allowed us to further strengthen this work. This resubmission took far longer than we had expected. Lab members who had completed experiments for the original submission continued their careers elsewhere, and our ability to recruit and train new members to address the points was impacted by the lockdown. We also faced substantial issues expanding our animal colony, which resulted from us moving institutions right before the lockdown. We hope the Reviewers can conclude that the revision period did not impact the relevance of our findings. One benefit was that key experimental results were replicated by different members of our group.

The Reviewers' points are shown below in italics, followed by our responses in blue. The revisions that address these points are also marked in blue in the manuscript text.

Reviewer 1

We are delighted that the Reviewer concludes that our study addresses “an important question” and “combines an impressive range of *in vivo* and *ex vivo* techniques” to provide “solid” data. We also thank her/him for the constructive points that guided us to perform new biochemical and immunostaining experiments, together with new analyses of immunostainings and electrophysiological data. These extensive revision experiments further strengthened this work. Our responses are below.

*1. The organization of the story is disjointed (going back and forth between the molecular composition of synapses and *in vivo* readouts in whole animals, then back to the ‘molecular remodeling’) making it hard to follow the message. It may be better to present the molecular changes/ synapse aberrations together, followed by the behavioral *in vivo* output of the synapse disruptions.*

We agree and have followed the Reviewer's advice. The revised manuscript now first presents the biochemical and synaptic data (Fig. 1, 2, 3), followed by *in vivo* analyses of activity (Fig. 4, 5), and behavior (Fig. 6, 7).

2. As the hypothesis is that this compensation is happening specifically in PFC, there are some key outstanding questions that should be answered to confirm this and support the existing data.

-An obvious question seems to be if there are significant differences in LRRTM1, SynCAM1, NLs and Nx in PFC synapses compared to other brain regions (i.e. are LRRTM1 and SynCAM1 particularly enriched there compared to hippocampus, whole forebrain etc.) If LRRTM1 was more highly expressed in PFC this would contribute to the argument that it might contribute to schizophrenia-like phenotypes.

We addressed this question in new biochemical experiments shown in Supplemental Figure 5. We compared the expression of α - and β -Neurexins, Neuroligin 1, LRRTM1, and SynCAM 1 in the PFC and hippocampus by quantitative immunoblotting, using WT mice at P55. Analysis of homogenates determined that LRRTM1 and SynCAM 1 are expressed at $54 \pm 7\%$ ($p=0.001$) and $18 \pm 5\%$ ($p=0.009$) higher levels in the hippocampus than PFC, respectively. This relatively lower abundance of LRRTM1 and SynCAM 1 in the PFC may provide a baseline from which their expression can be readily increased when needed, which is discussed on page 5, bottom. We hope the Reviewer agrees with this possibility.

-The mix of brain regions used for the biochemistry is inconsistent, making it difficult to make strong conclusions e.g. Fig 1. C, D use SPM from frontal cortex, but Supp. Fig 1A,B use SPM from forebrain. Due to the cooperativity of NL1 with LRRTM1, it is important to know if NL1 (and neurexins) are upregulated in SPM of frontal cortex.

We addressed the Reviewer's point regarding synaptic levels of Neuroligin-1 and Neurexins in the PFC rather than the frontal cortex to achieve high regional resolution. Our experiments used synaptosomes, which we could fractionate from the comparatively small PFC amounts using the protocol described in the Methods; PFC amounts were too low for SPM preparations, as expected. The new results show that alpha-Neurexins but not beta-Neurexins are more abundant in synaptosomes from LRRTM1 KO PFC (Figure 1c), consistent with the earlier SPM data from forebrain now shown in Supplemental Figure 1a. Neuroligin 1 was robustly detected at high levels in synaptosomes from LRRTM1 KO PFC (Figure 1d), again like in forebrain SPMs, with variability in the WT precluding us from determining whether this reached significance. SynCAM 1 was also significantly increased in synaptosomes from LRRTM1 KO PFC (Figure 1e), in agreement with the SPM data from frontal cortex. These new biochemical results support that loss of LRRTM 1 elevates synaptic alpha-Neurexins levels together with SynCAM 1 in the PFC.

We thank the Reviewer for pointing out that the use of multiple brain regions and fractions in this study makes the biochemical analyses difficult to follow. The main manuscript therefore only shows our new synaptosome data from the PFC in Figure 1c-e. All other fractionations are now provided in the Supplement, with SPMs from the total forebrain in Supplemental Figure 1a, SPMs from the frontal cortex in Supplemental Figures 1b, c, and SPMs from the hippocampus in Supplemental Figure 4a, b. To better communicate our biochemical experiments, we now also state in the Results the different advantages of these fractionations, with synaptosomes allowing for high brain regional resolution and SPMs yielding more highly purified synaptic plasma membranes but requiring larger tissue amounts.

3. The ex vivo imaging and analysis are nicely done (especially the use of machine learning to define a synapse), although they require further controls and there are some key experiments missing that would significantly strengthen the conclusions of the paper:

- Figure 1A: Synapse density is measured using the association of vGlut1 and Homer1 to define a synapse, revealing a strong reduction in excitatory synapses. The analysis is incomplete and needs to be expanded: are there changes in synapse area/volume? VGlut1 and homer should be analyzed independently for density/area, and the % of Vglut1 with/without homer (and the reverse) should be quantified (as reported in Fig 5 e,f- from the images it looks as if many Vglut1 positive synapses that do not have post-synaptic densities, and this is not shown in the data).

We thank the Reviewer for the positive comments on our immunohistochemical analyses. We extended our image quantification of mPFC immunostainings and analyzed:

- (i) Homer and vGlut1 puncta area, which are smaller in LRRTM1 KO PFC, see new Supplemental Figure 7;
- (ii) density of individual vGlut1 and Homer puncta, which showed that both are reduced in LRRTM1 KO mice and that SynCAM 1 KO mice have fewer Homer puncta, see new Figure 2e;
- (iii) the extent to which individual vGlut1 puncta co-localize with Homer puncta and vice versa, which are both reduced across all KO genotypes as shown in Figure 2g;
- (iv) the density of sites where vGlut1 and Homer puncta overlap, showing that they are reduced in all KO genotypes, see new Figure 2f.

As noted by the reviewer, 70% of vGlut1 puncta colocalize with Homer in our analysis of WT mice as shown in Figure 2d and quantified in Figure 2g, left panel. While a fraction of vGlut1 puncta appears isolated, we applied the rather restrictive approach of using single optical confocal sections instead of projection images for quantifications. We expect this to underreport apparent colocalization above and below the focal plane. An incomplete detection of postsynaptic sites could also contribute to a non-quantitative co-localization of vGlut1.

When we performed these new analyses, we double-checked all previous data and noted that a small subset of single KO data points had been erroneously copied when analyzing Homer co-localization with vGlut1. The data in the revised Figure 2g, right panel, show the corrected data. This earlier error was minor and did not alter the result or the interpretation of the DKO impairment.

-Considering Fig 1A forms part of the main crux of the story, analysis of other excitatory markers, such as PSD-95/AMPA, and for inhibitory synapses (particularly as E/I coordination and interneuron loss in this region are strongly implicated in Schizophrenia) should also be performed to get a sense of the extent of synapse disruption.

We performed new immunostainings for the inhibitory pre- and post-synaptic markers vGAT and Gephyrin in the mPFC. This determined that inhibitory synapse density is reduced in single LRRTM1 and SynCAM 1 KO mice as well as DKO mice, shown in the new Supplemental Figure 8a, b. In addition, the area of Gephyrin but not vGAT puncta is smaller across all KO genotypes as shown in the new Supplemental Figure 8c. As both LRRTM1 and SynCAM 1 are detected at excitatory synapses (Linhoff et al. *Neuron* 2009, PMID 19285470; Nozawa et al. *Neuron* 2022, PMID: 36007521; Loh et al. *Cell* 2016, PMID: 27565350; Perez de Arce et al. *Neuron* 2015, PMID 26687224), we discuss on page 7, second paragraph, that these effects could be compensatory responses to the loss of excitatory synapses in these KO mice. Our new data will allow addressing underlying mechanisms, including homeostatic responses, in future studies, which would go beyond this first report of cooperation between LRRTM1 and SynCAM 1. Regarding postsynaptic markers, the anti-Homer antibody provided in our hands the highest quality immunostaining in the mPFC for our machine learning-based puncta analysis. We hope the new inhibitory synapse analysis sufficiently addresses the Reviewer's point about synapse types.

-A direct comparison of synapse number between SynCAM1 and LRRTM1 single KO mice and the DKO mice is missing, although the imaging has been done (but analysis not included). Fig 5e,f reports Homer puncta colocalizing with vGLUT, but not synapse numbers. This would support the claim that KO of both proteins augments the effects on synapses.

We extended our analysis and show in the new Figure 2f that the density of excitatory synapses positive for both Homer and vGlut1 is reduced in SynCAM 1 KO PFC and even more so in LRRTM1 KO and DKO mice. The density of these immunohistochemically defined excitatory synapses is not further reduced in DKO PFC compared to LRRTM1 KO mice. As pointed out in the revised manuscript, this is unlike the spine data in Figure 2c showing a further density reduction in DKO mice compared to single KOs. We discuss on page 6, bottom, that the lack of a further loss of immunohistochemically detected excitatory synapses in the DKO may be due to a subset of vulnerable spines not being labeled by Homer. New electrophysiological data in Figure 4c now provide additional support for concerted excitatory impairments upon loss of both SynCAM 1 and LRRTM1. Please see point 5.

- The imaging of SynCAM1 in Supplementary Fig 1 C is unconvincing, and not quantified. This should be repeated as in Fig 1A with quantification and analysis of SynCAM1 with excitatory synaptic markers.

We have performed new immunohistochemical experiments to analyze SynCAM 1 staining in LRRTM1 KO PFC. The results are shown in Figure 1f, g. A fraction of SynCAM 1 staining was diffuse and non-synaptic, and we would have had to substantially threshold the images to analyze only puncta and their co-localization with excitatory markers. This would have excluded data, and we decided to analyze overall SynCAM 1 abundance. In agreement with our biochemical data in Figure 1e, quantification of these immunostainings determined that LRRTM1 loss increases SynCAM 1 amounts in the mPFC.

4. One of the conclusions of the paper is that ‘the adhesion molecules LRRTM1 and SynCAM 1 are interdependently expressed at synapses in the mPFC’. More imaging data are needed to fully support this statement. Although the biochemical data presented suggest this is the case, the presence of these proteins at synapses in the respective knockdown conditions should be analyzed. A key mechanistic question (acknowledged by the authors) is whether LRRTM1 and SynCAM1 are actually in the same synapses, and exactly how their actions are coordinated. Further, although NL1 upregulation in the LRRTM1 KO is mentioned, presumably NL1 is having a significant compensatory role in these mice- this needs to be addressed more fully (e.g. labeling with NL1 in the KO mice).

Regarding the expression changes, we now provide as stated above immunohistochemical data showing an increase in SynCAM 1 in the mPFC upon loss of LRRTM1 (Figure 1f, g). This agrees with our new biochemical data of synaptosomes prepared from the PFC (Figure 1e).

Despite our efforts to validate commercially available LRRTM1 antibodies in immunohistochemical staining applications, using KO tissue as control, we were not able to identify an antibody suitable to determine its localization. While this precluded us from addressing the Reviewer’s point directly, we would like to refer to Loh et al. *Cell* 2016 (PMID: 27565350). This study used HRP-tagged LRRTM1 as a proximity labeling reporter for the excitatory synaptic cleft proteome. This reporter robustly identified SynCAM 1 (listed under its gene name *Cadm1*), please see Loh et al., Figure 4A and Supplemental Table 1, Expt 1 and Expt 3. This proximity labeling approach has a very high ~20 nm spatial resolution due to the short lifetime of the enzyme-generated biotin radicals. These results hence provide evidence that SynCAM 1 is in close vicinity to LRRTM1 at the same synapses, which is now discussed on page 14, second paragraph. Once suitable antibodies detecting LRRTM1 in tissue are available, we will directly test this. Similarly, we could not perform immunohistochemical studies of Neuroligin 1 due to a lack of suitable antibodies. We agree that possible compensatory roles of Neuroligin-1 in the LRRTM1 KO are interesting and expect that this will be addressed in future studies.

5. The reduction of firing rate in of PFC layer II/III neurons is compelling, but mechanistic insight into what is driving this change in firing is lacking (and would link the alterations in synapses to the in vivo observations). Is it primarily due to the loss of post-synaptic sites/spines? Or is the the observed increase in NX cluster size causing problems with synaptic release? Is inhibition altered? Ex vivo electrophysiological analysis of the DKO mice would answer these questions.

Our group is unfortunately not equipped to perform slice physiological recordings of synaptic transmission, and we could not arrange a collaboration with a partner who could invest the effort for these recordings. We instead used our *in vivo* recordings to add neuron-type specific resolution to this study. We analyzed firing rates of regular spiking neurons, corresponding to excitatory neurons. Our new results, shown in Figure 4c, determined that DKO mice exhibit profoundly reduced spiking of regular spiking, excitatory neurons in mPFC compared to single KO mice or WT. This agrees with the DKO

loss of spines as a mechanism. Our interpretation follows studies that applied extracellular recordings from mouse PFC to investigate antipsychotics and NMDA antagonists (see for examples Jackson et al. *PNAS* 2004, PMID: 15159546; Kargieman et al. *PNAS* 2007, PMID: 17785415; Homayoun and Moghaddam *Biol Psychiatry* 2007, PMID: 17046721) and genetic mouse models (Alvarez et al. *J Neurosci* 2020, PMID: 32205341; Chini et al. *Neuron* 2020, PMID: 31733940).

We did not record enough events from fast-spiking inhibitory interneurons for their analysis. Instead, we measured the staining intensity for Parvalbumin, which is a correlate of the activity of fast-spiking interneurons as reported in Donato et al. *Nature* 2013, PMID: 24336286. As shown in the new Figure 4d, the loss of LRRTM1 increased the staining intensity for Parvalbumin. This agrees with the MUA firing rate increase in single LRRTM1 mice in Figure 4b. This phenotype was largely abrogated in DKO mice. This may be due to the fact that excitatory inputs to cortical PV-positive interneurons depend on SynCAM 1, which could contribute to silencing these cells in DKO mice (Ribic et al., *Cell Reports* 2019, PMID 30625321). Together, the MUA firing increase in the single LRRTM1 KO PFC may be driven by higher PV interneuron activity while the drop in overall MUA firing in the DKO could be due to the substantially fewer excitatory events due to spine loss of excitatory neurons together with lowered PV cell activity. We discuss these new physiological and immunostaining results on page 15, top, and hope the Reviewer can conclude that they add cellular resolution to better interpret these complex changes. We thank the Reviewer for this point as the new experiments provided more detailed physiological understanding and give further evidence that these two trans-synaptic systems cooperate.

We also performed new immunohistochemical stainings for Neurexins. These determined that the density of both total and vGlut1-positive synaptic Neurexin puncta is lower in DKO mPFC, replicating the data of the originally submitted manuscript that DKO mice had fewer total Neurexin puncta. Please see the new Figure 3c, d. The reduction of vGlut1-positive Neurexin puncta density in synapses of excitatory neurons we report is consistent with data from single and combinatorial deletion models of Neurexins, in which a reduction of spontaneous excitatory postsynaptic currents was observed (Missler et al. *Nature* 2003, PMID: 12827191; Zhang et al. *J Neurosci* 2005, PMID: 15858059; Etherton et al. *PNAS* 2009, PMID: 19822762). We did not measure an increase in the size of Neurexin puncta in these new experiments, unlike the data that had been obtained in our initially submitted work. This may be due to new antibody lots, altering staining intensity and thresholding. Due to this potential antibody-related variability, we decided not to include Neurexin puncta size analyses in the revised manuscript, which we hope the Reviewer agrees does not impact our conclusions.

6. Throughout, it is hard to reconcile the link between the initial findings of LRRTM1 loss in PFC through to the behavior with schizophrenia-like phenotypes. It is not surprising that substantially disrupting glutamatergic circuits in in the brain will produce those behavioral phenotypes (even if they are associated with the PFC). Currently, the authors do not provide compelling evidence linking LRRTM1 loss, SynCAM1 compensation and schizophrenia-like behaviors. Given the loose genetic association of LRRTM1 with schizophrenia, and no evidence to suggest that synapse loss specifically in PFC leads to the behaviors observed in these constitutive KO mice, the link with schizophrenia becomes somewhat tenuous. Addressing whether the PFC of DKO mice exhibit other cellular and circuit characteristics (eg. loss of PV+ interneurons) of schizophrenia pathology might go some way to addressing this issue.

We agree. The manuscript was revised to communicate our focus on synapse organization in the PFC and this region's importance for brain disorders, without suggesting a mechanistic link to schizophrenia.

Reviewer #2

We appreciate the Reviewer's detailed feedback and would like to thank her/him for concluding that this study shows "many consistent results to claim the concerted effects of LRRTM1 and SynCAM1 KOs on multiple levels from synapse level to behavioral one" and that this "is a strong point in this paper". We also appreciate the constructive criticisms and performed new analyses of electrophysiological data (point 1), new immunohistochemical analyses of inhibitory synaptic markers in the PFC (points 2, 7), and new biochemical experiments (points 3, 5). We hope the Reviewer agrees that these extensive additional data sets further strengthen the conclusions and aid the interpretation of the results. Please find our detailed responses below.

Major comments

1) This paper has no data regarding the assessment of synaptic functions. Although they performed fMRI imaging and in vivo recording, these allow them assess brain activity and neuronal activity but not directly assess the function of synapses. Due to the lack of the data about synapse functions, some anatomical data remain confusing for me to consider the author's claim about the concerted role of LRRTM1 and SynCAM1 on some behaviors. Especially, Fig.1a shows drastic reduction of excitatory synapses in the mPFC in the single KO of LRRTM1, and Fig. 5f shows significant reduction of Homer colocalization with VGLUT1 in the mPFC compared to WT littermates. However, LRRTM1 single KO mice show comparable behaviors to WT in all performed behavioral tests. One possible reason is that the remaining excitatory synapses in LRRTM1 KO mice would be more functional than those in WT littermates to compensate the reduction of excitatory synapse number. However, there is no electrophysiological data to explain such gap between anatomical deficits of excitatory synapses and no behavioral phenotypes. Further, Both LRRTM1 single KO mice and DKO mice show reduction of dendritic spines (Fig. 5a,b) and impaired colocalization of Homer with VGLUT1 but show opposite phenotypes in firing rate of mPFC pyramidal cells (Fig. 5d). This apparent discrepancy would be due to different phenotypes in synaptic functions between these two mouse lines. (e.g. again, synapses in LRRTM1 KO are more functional than those in WT due to SynCAM1 upregulation, whereas synapses in DKO are less functional than in WT because of loss of the SynCAM1 upregulation as well as its default expression.). Thus, the assessment of synapse function by electrophysiological experiments is indispensable to achieve more convincing data interpretation from molecular, anatomical level to behavioral level.

As summarized by the Reviewer, this work focuses on molecular and network-level changes that are expanded by our in vivo recordings but does not analyze synaptic physiology. We are not equipped to perform slice physiology and could not arrange a collaboration partner who was able to invest the effort for these recordings. While we could not test whether synapses in the DKO PFC may be less functional than in WT PFC, we include this idea in the revised Discussion. Using our available resources, we instead responded to the Reviewer's point by adding neuron-type specific results and analyzed single unit events in the *in vivo* recordings. Our new results in Figure 4c show in DKO mice a substantial reduction of the spike frequency of regular spiking neurons, corresponding to excitatory neurons, compared to single LRRTM1 and SynCAM 1 KO and WT mice. These data are consistent with the strong loss of morphologically defined excitatory synapses in the DKO PFC.

We would like to apologize that we did not state that the multi-unit activity now shown in Figure 4b represents an aggregate signal of all local spiking activity of both excitatory and inhibitory events. The Reviewer's point that there is an apparent discrepancy between the increased MUA firing in the LRRTM1

KO and the decreased rate in the DKO could therefore be explained if interneuron activity differs between these genotypes. Our physiological measurements did not allow testing this as we did not record enough of the less abundant fast spiking units. We instead measured the staining intensity for Parvalbumin which is a correlate of the activity of fast-spiking interneurons as reported in Donato et al. *Nature* 2013, PMID: 24336286. Indeed, our new results in Figure 4d show that the single loss of LRRTM1 increased the intensity of Parvalbumin staining. This supports a higher activity of Parvalbumin-positive interneurons in LRRTM1 KO PFC that would add firing events to increase overall MUA rates. This phenotype is abrogated when both LRRTM1 and SynCAM 1 are absent. The fact that excitatory inputs to PV-positive interneurons depend in part on SynCAM 1 (Ribic et al., *Cell Reports* 2019, PMID 30625321) could contribute to a silencing of these cells in DKO mice. The MUA firing increase in the single LRRTM1 KO PFC may be driven by higher PV cell activity while the lower overall MUA firing in the DKO could be due to the substantially fewer excitatory events after spine loss of excitatory neurons together with lowered PV cell activity. We discuss these results that provide new evidence that these two trans-synaptic systems cooperate on page 15, top.

It can be considered that the extent of synapse loss in LRRTM1 KO mice remains above a threshold impairing the behaviors measured in this study and that the additional loss of synapses in LRRTM1/SynCAM 1 DKO mice causes synapse numbers to drop below that threshold. This is now discussed on page 15, second paragraph.

2) Excitation/inhibition (E/I) imbalance in the mPFC is known to be one of key pathophysiological conditions underlying social behavioral deficits. However, in this paper, the authors have not analyzed anything regarding inhibitory synapse. Indeed, previous studies have suggested that LRRTM1 and SynCAM1 represent excitatory, but not inhibitory, synapse localization and trigger excitatory, but not inhibitory, synaptogenesis. However, the reduction of excitatory synapses primarily induced by LRRTM1 single KO and LRRTM1/SynCAM1 DKO might result in some changes in GABAergic inhibitory synapse number and/or function as a secondary effect presumably through activity-dependent homeostatic mechanism and/or compensatory mechanism that normalizes the E/I imbalance. Further, the author's data show LRRTM1 KO-induced upregulation of synaptic alpha-neurexins, which have synaptogenic activity to induce inhibitory, but not excitatory, postsynaptic differentiation (J Biol Chem (2008), 283:2323-34). The further analysis of inhibitory synapses to elucidate whether E/I imbalance in the mPFC occurs or not in each KO mouse line is essential to improve data interpretation of Fig.3 c.d (Social behavioral test) and Fig 4 (Brain activity and neural activity phenotypes).

As stated by the Reviewer, we focused in this study on excitatory synapses as both LRRTM1 and SynCAM 1 localize to them (Linhoff et al. *Neuron* 2009, PMID 19285470; Nozawa et al. *Neuron* 2022, PMID: 36007521; Loh et al. *Cell* 2016, PMID: 27565350; Perez de Arce et al. *Neuron* 2015, PMID 26687224). We addressed the Reviewer's question and extended this work by performing new immunostainings for inhibitory synaptic markers. This determined that the density of inhibitory synapses in the mPFC is reduced in all KO genotypes, as shown in the new Supplemental Figure 8. Our results agree with the Reviewer's notion, and we discuss on page 7, second paragraph, that these effects could be responses to the loss of excitatory synapses in the single and double KO mice. Our new data will allow testing underlying mechanisms in future studies, including possible homeostatic responses and changes in the strength of the remaining excitatory and inhibitory synapses, which would go beyond this first report of cooperation between LRRTM1 and SynCAM 1.

Regarding the behavioral impact, we now discuss on page 15, second paragraph, that the phenotypes may reflect impairments in connectivity or alterations in E/I balance that are not sufficiently compensated in DKO mice.

With respect to Neurexins, one possibility for why their elevation did not increase inhibitory postsynaptic sites is that this role of α -Neurexins could be region specific as our studies were performed in the cortex, unlike the analyses of cultured hippocampal neurons in Kang et al. *J Biol Chem* 2008, PMID: 18006501, or because of counteracting regulatory mechanisms *in vivo* that are not at play *in vitro*.

3) This paper has no experimental evidence regarding mechanisms underlying interdependent synaptic expression of LRRTM1/SynCAM1 in the mPFC. Therefore, the current paper is relatively descriptive with just listing many phenotypes of single KO and DKO mice. Also, considering several previous studies has already shown interdependent expression and/or concerted effects of two different types of synaptic organizers with providing mechanistic insight (e.g. *Slitrk3/NLG2 in Neuron* (2017), 96:808-826, *LRRTMs/NLG1/3 in PNAS* (2011), 108: 16502-9 and *J Cell Biol* (2011), 194(2):323-34), I think that it would be necessary for this paper to add some experimental data that can provide mechanistic insight into how LRRTM1 and SynCAM1 interdependently regulate the expression of each other in the mPFC.

In the discussion section, the authors discuss a slot model as one of possible mechanisms. It has been reported that presynaptic scaffold CASK binds to SynCAM1 and neurexin, an LRRTM1-binding partner. Thus, it would be interesting to test whether SynCAM1 and LRRTM1 compete with each other for CASK binding and whether the loss of one increases scaffold slots available for the other by losing the binding competition. Especially, given that the authors show altered expression and puncta size of neurexin in the mPFC of the DKO mice, it would be crucial for this paper to characterize CASK, a common binding partner for neurexin (LRRTM1-binding partner) and SynCAM1.

In addition, is it possible whether SynCAM1 can bind to neurexin itself? If yes, is it possible whether SynCAM1 and LRRTM1 compete with each other for neurexin-binding? The authors mention that SynCAM1 does not bind LRRTM1 (refs 36,37), but there is no information or discussion about SynCAM1/Neurexin binding.

We appreciate these suggestions. With respect to CASK, we tested in new experiments whether the changes in LRRTM1 KO synaptosomes from the PFC extend to CASK, which would point to it being part of compensatory responses. Our quantification provided no evidence for significant changes in synaptic CASK upon LRRTM1 loss, please see the panel (n=4 WT, 3 LRRTM1 KO mice per genotype).

Regarding a potential competition, the PDZ domain interaction motifs of SynCAM 1 and LRRTM1 differ, and only SynCAM 1 is predicted to bind CASK as confirmed in the initial characterization of SynCAM 1. We therefore did not further pursue a potential role of CASK in LRRTM1-dependent compensatory changes for this revision. The Reviewer's point that scaffold molecules may change in the KO mice to control the availability of membrane slots is interesting, and we aim to pursue it for other proteins.

We addressed the Reviewer's question of whether SynCAM 1 can bind to Neurexin applying co-IP from heterologously expressing HEK293 cells, shown in the new Supplemental Figure 2. We validated the expected direct interaction of Neurexin 1 α with Neuroligin 1. In the same experiments, we did not detect an interaction between Neurexin 1 α and SynCAM 1. Our findings agree with an interaction

screen of adhesion molecules that did not find that Neurexin 1 α and SynCAM 1 bind each other (Ranaivoson et al. *Structure* 2019, PMID: 30956130). These results provide evidence against a model that SynCAM 1 and LRRTM1 compete with each other for Neurexin binding.

We hope the Reviewer agrees that these new experiments have tested a first set of mechanistic ideas that can be pursued in future studies.

We would also like to thank the Reviewer for pointing out the relevance of the study of Slitrk2/Nlgn2 by Li and colleagues in *Neuron* 2017 for our work. It is now cited in the Introduction on page 2.

Minor comments:

4) *Although the authors wrote that “SynCAM1 is predominantly postsynaptic (line 114)”, the phrase of “the postsynaptic adhesion molecules LRRTM1 and SynCAM1” in line 394-395 should be an over-statement and biased because SynCAM1 is also expressed at presynaptic site to act as a presynaptic adhesion molecule with homophilic trans-interaction with postsynaptic SynCAM1. Also, the authors use global KO of SynCAM1, which cannot allow them to discriminate the role of postsynaptic SynCAM from that of presynaptic one. More careful data interpretation and unbiased discussion are necessary.*

We agree and thank the Reviewer for pointing out this statement. We removed statements that could indicate that the reported effects involve only a postsynaptic locus and now state that the presynaptic population of SynCAM 1 may be involved in trans-synaptic cooperation with LRRTM1, see page 14, top.

5) *The authors use anti-neurexin antibody (Millipore, ABN161), which can recognize both alpha- and beta-neurexins. However, they quantified only alpha-neurexin. They need to quantify beta-neurexins too.*

We optimized our fractionation protocol to obtain synaptosomes from the comparatively small PFC tissue amounts to address the Reviewer’s question. Our new immunoblotting data in Figure 1c use the Millipore ABN161 antibody and show that α -Neurexins are increased in this fraction in LRRTM1 KO mice compared to WT. We did not measure changes in β -Neurexins. β -Neurexin bands were quantified between 60-100 kDa based on the loss of immunoblot signal in this molecular weight range in triple Neurexin KO mice (Uemura et al. *Cell Rep* 2022, PMID: 35385735), the detection of β -Neurexins in beta-isoform specific KO/rescue mice (Klatt et al. *Cell Reports* 2021, PMID: 34133920), as well as the ligand-based enrichment of β -Neurexin isoforms (Roppongi et al. *Neuron* 2020, PMID: 31995730). Our results indicate that α -isoform specific properties modulate the Neurexin increase in absence of LRRTM1.

6) *How about synaptic expression of alpha- and beta-neurexins and neuroligin-1 in DKO mice? These results would be important to discuss whether compensatory role of SynCAM1 under LRRTM1 deletion is dependent on or independent from these synaptogenic molecules and would be also helpful to consider possible mechanisms underlying interdependent synaptic expression of LRRTM1/SynCAM1 in the mPFC.*

We encountered significant breeding issues with this DKO line during the lockdowns, while we were re-establishing our mouse lines following our move. While we were able to obtain DKO mice for immunohistochemical analyses to address point 7, and see the new Figure 3, ongoing issues precluded us from obtaining a sufficient number of DKO mice for biochemical studies. We hope the Reviewer understands our decision to prioritize the immunohistochemical experiments.

7) In Fig. 6b, the representative image of PFC SynCAM1 KO shows the weakest neurexin immunoreactivity, which is not consistent with the quantitative data. It should be replaced with a better representative one.

In addition, it is unclear what the enlarged neurexin puncta size in the DKO mice mean, enlarged presynaptic terminals, an increase in extra-synaptic neurexin puncta, or anything else? To answer this question, co-immunolabeling of presynaptic marker (e.g. Bassoon or Vglut1) with neurexin like Fig 6a and the quantification of puncta density and size of synaptic neurexin and non-synaptic neurexin would be necessary.

We performed extensive new experiments to address this point through co-immunostaining for Neurexins using the Millipore ABN161 antibody and for vGlut1. The data obtained in new cohorts of mice show that approx. 30% of Neurexin puncta in PFC layer II/III in WT colocalize with this excitatory presynaptic marker, please see Figure 3b-d. This agrees with Trotter et al., 2019, *J Cell Biol* PMID: 31262725 and Klatt et al. *Cell Reports* 2021, PMID: 34133920 that showed that no more than 40% of synaptic terminals contain α - and β -Neurexins. Notably, the density of both total (Fig. 3c) and synaptic (Fig. 3d) Neurexin puncta was reduced in DKO compared to LRRTM1 PFC. This reproduced and extended the result in the originally submitted manuscript that showed fewer total Neurexin puncta in DKO compared to LRRTM1 KO. The results are consistent with single and combinatorial deletion mouse models of Neurexins with impaired excitatory transmission (Missler et al. *Nature* 2003, PMID: 12827191; Zhang et al. *J Neurosci* 2005, PMID: 15858059; Etherton et al. *PNAS* 2009, PMID: 19822762). We found this decrease in the density of total and vGlut1-positive Neurexin puncta in the DKO we determined in response to the Reviewer's question very informative and thank the Reviewer for this point.

We also analyzed Neurexin puncta size in the immunostainings from the new cohorts. The results did not replicate the originally observed subtle area increase. We analyzed mice of the same age and the same PFC areas, albeit bred in a different facility, and the reason for this different result may be due to new antibody lots and staining procedures, which could alter staining intensity and in turn thresholding. As the earlier analysis of Neurexin puncta area did not replicate across independently analyzed cohorts, we removed it, which does not alter key conclusions from this study.

8) The authors used parametric tests (e.g. student's t-test for two groups and ANOVA for multiple groups) for all statistical except Fig 4d but used Kruskal-Wallis ANOVA (non-parametric test) for only Fig.4d. However, there is no description about the criteria in which parametric or non-parametric test was used (e.g. assumption of normality). In the statics section in the Reporting Summary, they selected "N/A" in "A description of any assumptions or corrections, such as tests of normality and adjustment for multiple comparisons", suggesting that they have not tried assumption of normality even when they used a parametric test. However, if the data are not normally distributed, using parametric test is not appropriate and thereby results in misleading conditions. Therefore, the authors need to revalidate if they actually use an appropriate statistical test for each data set by assuming data normality and homoscedasticity.

The Kruskal-Wallis test was performed for the data now shown in Figure 4b because MUA firing rates were the only data sets in our study not normally distributed in all groups. Specifically, there appeared to be two populations in the KO lines, while WT data was normally distributed. This non-normal distribution of firing rates could reflect that the loss of LRRTM1 and SynCAM 1 has more pronounced phenotypes in distinct cell types. This would be consistent with their cell-type specific roles in synapse

specification as reported for LRRTM1 (e.g. Dhumeet et al. *Elife* 2022, PMID: 35662394; Schroeder et al., *Neuron* 2018, PMID 29983322) and SynCAM 1 (Ribic et al., *Cell Reports* 2019, PMID 30625321). The non-normality may arise from variability in pyramidal neurons in layer 2/3 targeted in the recordings, which is a heterogeneous population of neurons in the PFC marked by diverse inputs/outputs. This is speculative and we are not discussing this in the manuscript. The above statistical information is now provided in the Methods section ‘Data and statistical analysis’, and the Reporting Summary was revised.

Reviewer #3

We thank the Reviewer for concluding that this work addresses an “important issue”, and that “the experimental approach and techniques used in this paper were impeccably conducted” and “rigorously designed”. We are delighted about the conclusion that “this paper provides exciting new information” about synapse development. The Reviewer’s specific points were very helpful, and we provide additional information and new experimental data as described below.

1. Data in Figure 1 showing LRRTM1 control of excitatory synapse density and expression should report the exact P values for significant differences among the genotypes, rather than $p < 0.05$.

We now report the exact p values and additional statistical information for the immunostainings in Figures 1b, 2e-g, 3c, d, and Supplemental Figures 7 and 8. This information is provided in the revised figure legends and where numerical differences are stated in the Results.

2. Tests of social behavior as shown in Figure 3 demonstrate that DKO mice are impaired for sociability in the 3 chambered test for object versus stranger mouse. For a more rigorous second order test, the authors should consider adding the more complex 3 chambered test for social novelty preference involving a second unfamiliar mouse.

We performed the suggested experiments. Using the experimental setup in our facility, we found that mice of all genotypes exhibited a trend toward more interaction with the unfamiliar mouse as shown below. However, this difference did not reach significance for any genotype, including WT, please see the figure.

The number of mice we analyzed was in a range we consider acceptable for these experiments (n= 15 WT, 16 LRRTM1 KO, 15 SynCAM 1 KO, 12 DKO). We currently

do not know why we did not observe a pronounced preference for the unfamiliar mouse in our WT controls and did not include these new, inconclusive results. We hope that the Reviewer can conclude that the test of object versus stranger mouse is satisfactorily informative to assess sociability in the context of our work.

3. In the electrophysiology experiments of Figure 4 c,d, was a Kruskal-Wallis Anova performed because there were 2 populations of firing rates within the mutant data? If so, what would be an explanation for the different populations?

The Kruskal-Wallis test was performed for the MUA firing rates now shown in Figure 4b because the data was not normally distributed in all groups. There appeared to be two populations in the KO lines. WT data was normally distributed. This non-normal distribution of firing rate data in the KO mice could reflect that the loss of LRRTM1 and SynCAM 1 has a more pronounced phenotype in distinct cell types. This would be consistent with their cell-type specific roles in synapse specification as reported previously for LRRTM1 (e.g. Dhumeet al. *Elife* 2022, PMID: 35662394; Schroeder et al., *Neuron* 2018, PMID 29983322) and SynCAM 1 (Ribic et al., *Cell Reports* 2019, PMID 30625321). The above information is now provided in the Methods section 'Data and statistical analysis'.

We cannot determine from the current dataset the mPFC neuron type(s) whose firing is specifically altered in the KOs. We consider it likely that this non-normality arises from effects in pyramidal neurons in layer 2/3 since that was targeted with the recordings, which is a heterogeneous population of neurons in the PFC marked by diverse inputs/outputs. As this is speculative, we are not discussing this in the manuscript. Studies such as *in vivo* optical physiology recordings could address this.

4. In the spine density results of Figure 5 explain why there is a separate category for large head mushroom spines, as generally these are included within the standard mushroom spine category. Since spines dynamically "morph" among the different morphologies, mushroom, stubby and thin spines are not truly independent variables. Thus, Anova may not be applicable; instead multinomial regression analysis would be more appropriate for comparing differences among genotypes. The authors should consult with a biostatistician about this. Minor point, density is misspelled on the y-axis of b and d.

We would like to apologize that we used non-standard terms in the original submission, where we referred to the standard mushroom spine category as "large head mushroom spines" and to thin spines as "mushroom spines". The terms we now use for spine types in the revised manuscript follow Harris et al. *J Neuroscience* 1992, PMID: 1613552 and we added a diagram in Figure 2a illustrating our classification criteria. These are also stated in the Methods.

We would like to thank the Reviewer for her/his guidance on the statistical analysis of spine type distributions across genotypes. Originally, a two-way ANOVA was performed for morphologies and genotypes. We agree that the four types of dendritic protrusions are not independent. As suggested by the Reviewer, we consulted with a research scientist in biostatistics at the Yale School of Medicine, Dr. Feng Dai. Dr. Dai reviewed the data and recommended performing a one-way (genotype) ANOVA separately for each dendritic protrusion type (thin, mushroom, and stubby spines, and filopodia-like), followed by pairwise comparisons and the use of Tukey's test to detect differences. We re-analyzed the data accordingly and show the statistical significance of differences for each type of dendritic protrusion in the revised Figure 2c and 2i. This corrected statistical analysis supports that DKO mice exhibit a significant, non-linear loss of thin spines compared to single KO mice. The above information about our statistical analysis of these data is now provided in the Methods section 'Data and statistical analysis'.

We corrected the misspelling in the figure and appreciate the careful reading.

REVIEWER COMMENTS

Reviewer #1 (Remarks to the Author):

To their credit, the authors have completed a sizeable body of experiments to address my concerns, in what sounds like a challenging couple of years. However, without data on synaptic function linking the molecular observations in Figures 1-3 to the in vivo recording data in Figs. 4-5, my enthusiasm for the manuscript is still reduced. In addition, the mechanistic evidence for a 'concerted' role for these adhesion molecules is still fairly weak. Overall, the paper is still mostly observational with little true mechanistic insight, and in my opinion, functional readouts of synaptic function is crucial for publication in Nature Communications.

Reviewer #2 (Remarks to the Author):

The authors very carefully addressed almost all of my major and minor comments, therefore I think that the current manuscript has been significantly improved and enough to support their conclusion about concerted functions of LRRTM1 and SynCAM1 for regulating synapse organization, neuronal activity and brain functions although I still have only two minor comments.

1) In Fig. 2c, there is no labeling of mouse genotypes on each bar graph.

2) Regarding a potential competition for CASK binding, I actually wanted to see whether SynCAM1 and neurexin (LRRTM1-binding partner) compete with each other for CASK binding although I miss-wrote as my review comment that "Thus, it would be interesting to test whether SynCAM1 and LRRTM1 compete with each other for CASK binding". If the authors can easily try a competitive binding assay for SynCAM1 and neurexin on CASK, it will be recommended. However, it would be not absolutely necessary because the authors put a discussion about possible underlying mechanisms at page 14 line 358-362 and because they did two additional experiments to answer my comment by investigating CASK expression in PFC synaptosome and by checking no interaction between neurexin and SynCAM1.

Reviewer #3 (Remarks to the Author):

The manuscript NCOMMS-20-39222A from the Biederer laboratory has been revised satisfactorily in regard to all of my prior comments.

I think it is an outstanding contribution to the field of cell adhesion molecules action in neural development.

Response to the Reviewers' Comments on the revised manuscript NCOMMS-20-39222

Reviewer 1:

To their credit, the authors have completed a sizeable body of experiments to address my concerns, in what sounds like a challenging couple of years. However, without data on synaptic function linking the molecular observations in Figures 1-3 to the in vivo recording data in Figs. 4-5, my enthusiasm for the manuscript is still reduced. In addition, the mechanistic evidence for a 'concerted' role for these adhesion molecules is still fairly weak. Overall, the paper is still mostly observational with little true mechanistic insight, and in my opinion, functional readouts of synaptic function is crucial for publication in Nature Communications.

We are pleased that the extensive new experiments and analyses performed for the revision, including functional *in vivo* recording data, addressed a significant number of points the Reviewer raised. This being the first report on the intriguing phenotypes of LRRTM1/SynCAM 1 DKO mice, we agree that future studies can now extend mechanistic insights.

Reviewer 2:

The authors very carefully addressed almost all of my major and minor comments, therefore I think that the current manuscript has been significantly improved and enough to support their conclusion about concerted functions of LRRTM1 and SynCAM1 for regulating synapse organization, neuronal activity and brain functions although I still have only two minor comments.

We thank the Reviewer for the positive comments on our revised study and are delighted that she/he concluded that our revision was careful.

1) In Fig. 2c, there is no labeling of mouse genotypes on each bar graph.

We thank the Reviewer for noting this and have added the legend for the mouse genotypes.

2) Regarding a potential competition for CASK binding, I actually wanted to see whether SynCAM1 and neurexin (LRRTM1-binding partner) compete with each other for CASK binding although I miss-wrote as my review comment that "Thus, it would be interesting to test whether SynCAM1 and LRRTM1 compete with each other for CASK binding". If the authors can easily try a competitive binding assay for SynCAM1 and neurexin on CASK, it will be recommended. However, it would be not absolutely necessary because the authors put a discussion about possible underlying mechanisms at page 14 line 358-362 and because they did two additional experiments to answer my comment by investigating CASK expression in PFC synaptosome and by checking no interaction between neurexin and SynCAM1.

We are pleased that the experiments and discussion points added in the revision process addressed several questions the Reviewer raised. We apologize for the misunderstanding of the one question about CASK binding. We considered setting up new experiments including biochemical approaches with purified proteins but would unfortunately not be able to complete these for a timely revision of the final manuscript. We hope that the additions for the revision and the new data presented in the original response to the Reviewer are sufficient to support our conclusions.

Reviewer 3:

The manuscript NCOMMS-20-39222A from the Biederer laboratory has been revised satisfactorily in regard to all of my prior comments.

I think it is an outstanding contribution to the field of cell adhesion molecules action in neural development.

We thank the Reviewer for her/his strong support of this revised study and the rationale motivating this work.